# Neuromem: A Granular Decomposition of the Streaming Lifecycle in External Memory for LLMs

**Ruicheng Zhang** [1]  **Xinyi Li** [1]  **Tianyi Xu** [1]  **Shuhao Zhang** [1]  **Xiaofei Liao** [1]  **Hai Jin** [1]

## Abstract

Most evaluations of *External Memory Module* assume a static setting: memory is built offline and queried at a fixed state. In practice, memory is *streaming*: new facts arrive continuously, insertions interleave with retrievals, and the memory state evolves while the model is serving queries. In this regime, accuracy and cost are governed by the full memory lifecycle, which encompasses the ingestion, maintenance, retrieval, and integration of information into generation. We present *Neuromem*, a scalable testbed that benchmarks *External Memory Modules* under an interleaved insertion-and-retrieval protocol and decomposes its lifecycle into five dimensions including *memory data structure*, *normalization strategy*, *consolidation policy*, *query formulation strategy*, and *context integration mechanism*. Using three representative datasets LoCoMo, LONG-MEMEVAL, and MEMORYAGENTBENCH, *Neuromem* evaluates interchangeable variants within a shared serving stack, reporting token-level F1 and insertion/retrieval latency. Overall, we observe that performance typically degrades as memory grows across rounds, and time-related queries remain the most challenging category. The memory data structure largely determines the attainable quality frontier, while aggressive compression and generative integration mechanisms mostly shift cost between insertion and retrieval with limited accuracy gain.

[1]National Engineering Research Center for Big Data Technology and System, Service Computing Technology and System Lab, Cluster and Grid Computing Lab, School of Computer Science and Technology, Huazhong University of Science and Technology, Wuhan, China. Correspondence to: Shuhao Zhang <shuhao_zhang@hust.edu.cn>.

*Proceedings of the 43rd International Conference on Machine Learning*, Seoul, South Korea. PMLR 306, 2026. Copyright 2026 by the author(s).

## 1. Introduction

*External Memory Modules* are becoming a core building block for long-horizon, interactive LLM systems, including medical assistants (Yuan et al., 2024), embodied agents (Liang et al., 2025; Salama et al., 2025; Anonymous, 2026), and educational tutoring (Pan et al., 2025; Huang et al., 2024; Salemi et al., 2024). By persisting, updating, and selectively retrieving information beyond the context window, *External Memory Modules* provide access to relevant prior facts during reasoning and enable cross-session continuity. Unlike *parametric* memory (Behrouz et al., 2025; Yan et al., 2026; Yu et al., 2026), where static knowledge is integrated into model weights via fine-tuning or continual learning, *External Memory Modules* maintain state in an explicit, editable store decoupled from the backbone model. While this separation enables continual updates without model modification, it transforms memory into a complex system component requiring insertion and retrieval pipelines designed under strict latency constraints.

Today, driven by diverse application demands and deployment budgets, *External Memory Modules* have expanded into a large and fast-evolving design space (Madaan et al., 2022; Dalvi Mishra et al., 2022; Huang et al., 2024; Gutiérrez et al., 2024; Ong et al., 2025; Xu et al., 2025; Latimer et al., 2025; Li et al., 2025; Huang et al., 2025; Chen et al., 2026; Tao et al., 2026). Yet, choosing an appropriate design for a concrete workload remains difficult. Furthermore, memory is *streaming* in practice: new facts arrive continuously, insertions interleave with retrievals, and the memory state evolves while the model is serving queries (Chandrasekaran & Franklin, 2002). This interleaving exposes non-obvious trade-offs and reveals how computational costs are shifted between insertion and retrieval (Zhang et al., 2025), but existing evaluations (Maharana et al., 2024; Wu et al., 2025; Tan et al., 2025) rarely make it clear which design decisions along the memory lifecycle drive the observed accuracy–latency outcomes.

Recent benchmark efforts have begun to move beyond single-number leaderboards. LoCoMo and MemBench (Maharana et al., 2024; Tan et al., 2025) emphasize long-horizon interactions and introduce efficiency measurements, while Minerva (Xia et al., 2025) takes a complementary angle:

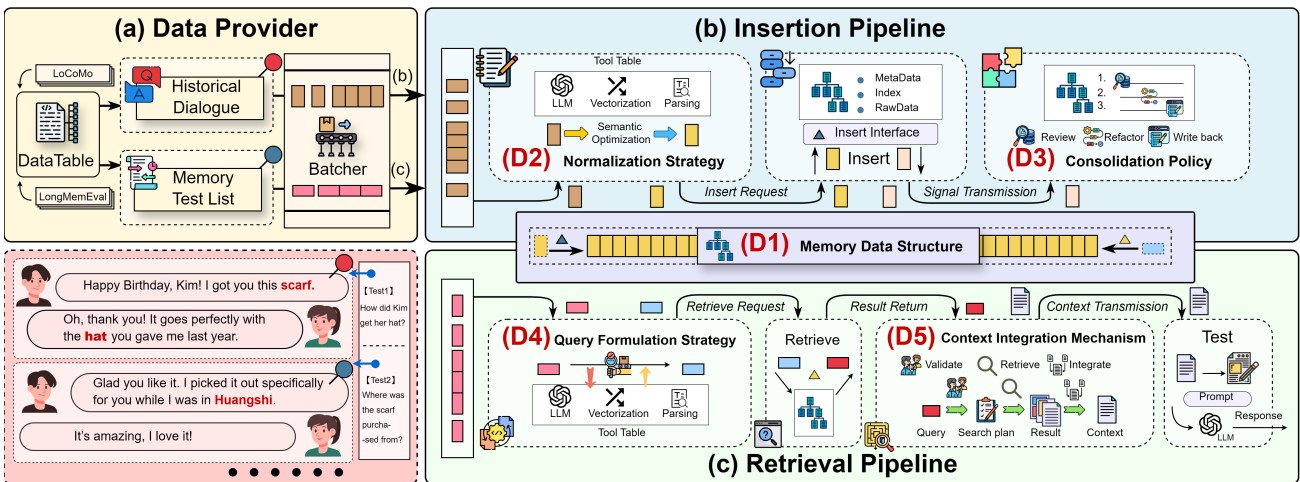

*Figure 1.* Overview of *Neuromem*. We decompose the lifecycle into five dimensions anchored by *Memory Data Structure* (D1). The workflow spans the (b) **Insertion Pipeline**, including *Normalization Strategy* (D2) and *Consolidation Policy* (D3), and the (c) **Retrieval Pipeline**, including *Query Formulation Strategy* (D4) and *Context Integration Mechanism* (D5), enabling controlled ablations under an interleaved insertion and retrieval protocol.

it programmatically generates fine-grained, interpretable *in-context* memory tests to diagnose what language models can do with their prompt memory. These are valuable foundations, but critical gaps remain for understanding *External Memory Modules* in realistic deployments. First, many comparisons remain coarse-grained and end-to-end, ranking whole systems and obscuring the impact of specific design decisions. Second, efficiency reporting is often under-specified: average read/write times conceal where overhead is incurred, and system-level metrics (e.g., memory foot-print) and attribution between memory-module overhead and model inference are frequently missing. Third, many evaluations still operate in a static or quasi-static regime (offline build, then query), whereas real deployments are dominated by interleaved insertion and retrieval, where interference accumulates as memory grows and maintenance competes with serving latency.

To capture these dynamics, we argue that evaluation must be both *streaming* and *lifecycle-aware*. Streaming evaluation reflects the interleaved insertion and retrieval regime in which memory evolves over time. Complementarily, lifecycle-aware evaluation addresses the attribution challenge by dissecting the monolithic memory system into granular stages. To operationalize this, we decompose *External Memory Modules* into five dimensions covering the full data lifecycle: *Memory Data Structure* (D1), *Normalization Strategy* (D2), *Consolidation Policy* (D3), *Query Formulation Strategy* (D4), and *Context Integration Mechanism* (D5) (Figure 1).

We instantiate this framework in *Neuromem*, a scalable testbed that benchmarks *External Memory Modules* under the interleaved insertion and retrieval protocol. By isolat-

ing design dimensions within a shared serving stack, *Neuromem* enables precise attribution of performance gains. Using three representative datasets LoCoMo, LONG-MEMEVAL and MEMORYAGENTBENCH, *Neuromem* evaluates interchangeable variants, reporting token-level F1 together with insertion/retrieval latency.

Our analysis suggests a recurring cost-shifting pattern: systems often relocate computational debt between insertion and retrieval rather than eliminating it. We find that hybrid data structures (D1) largely shape the observed quality frontier, while expensive operations like summarization and multi-query fusion often introduce substantial latency for limited gains under the tested setup. Finally, performance degrades as memory accumulates, with temporal reasoning remaining a persistent bottleneck.

**Contributions.** This paper makes three contributions. (1) We introduce *Neuromem*, an open-source testbed for evaluating *External Memory Modules* under a streaming protocol within a unified serving stack. (2) We propose a granular decomposition of memory architectures into five design dimensions and implement representative design variants for each, enabling precise, controlled ablations. (3) We conduct an extensive empirical study across diverse long-horizon benchmarks to distill practical design guidelines on balancing reasoning quality with insertion and retrieval costs.

**Paper organization.** Section 2 reviews related work on external memory modules and evaluation benchmarks. Section 3 formalizes the problem setting and notation, and decomposes the memory lifecycle into five design dimensions (D1–D5). Section 4 introduces the interleaved streaming evaluation protocol, details the system instantiations mapped to our taxonomy, and describes the metrics and

experimental platform. Section 5 presents the experimental analysis and key findings, and Section 6 concludes with implications and future directions.

## 2. Related Work

### 2.1. External Memory Modules

To address the fundamental limitations of **statelessness** and **finite context windows**, which confine models to ephemeral interactions and thereby preclude long-horizon coherence, personalized adaptation, and continuous self-evolution, a wide array of *External Memory Modules* has been proposed. These systems serve as a bridge for long-horizon continuity, ranging from general-purpose retrieval tools (Kang et al., 2025; Zhong et al., 2024) to specialized agentic frameworks (Zhang et al., 2024; Liu et al., 2025); a comprehensive review is provided in Appendix A.1.1. Despite this diversity, current designs remain predominantly *monolithic*: they tightly couple storage with maintenance logic into "black box" systems, obscuring the contribution of individual lifecycle stages and hindering optimization for real-time interaction.

### 2.2. Evaluation Benchmarks

Parallel to the rapid evolution of *External Memory Modules*, the evaluation landscape has made significant strides in expanding task diversity and reasoning complexity, as exemplified by LOCOMO, LONGMEMEVAL, and MEMORYAGENTBENCH (see Appendix A.1.2). Complementing these accuracy-focused benchmarks, pioneering efforts like MEMBENCH have also made valuable contributions by incorporating efficiency metrics into the evaluation loop. However, even these advancements are limited by their reliance on isolated latency measurements and static, retrospective protocols; this inability to capture the interleaved dynamics of continuous state evolution motivates our proposal for a lifecycle-aware streaming evaluation.

## 3. Design of External Memory Module

### 3.1. Problem Setting

We formalize the *External Memory Module* as a stateful component processing a continuous stream of operations. Let $\mathcal{R} = \{r_i = (\tau_i, \text{TYPE}_i, \text{PAYLOAD}_i)\}_{i=1}^{\infty}$ denote the request stream, where each request comprises a timestamp $\tau_i$, an operation type $\text{TYPE}_i \in \{\text{INSERT}, \text{RETRIEVE}\}$, and a payload. The payload is denoted as context $h$ for INSERT operations and query $q$ for RETRIEVE operations.

The memory maintains a state sequence $\{M^{(k)}\}_{k=0}^{\infty}$ and manages its lifecycle via two primary pipelines:

**Insertion Pipeline.** The state $M^{(k)}$ is derived from the previous state by processing the $k$-th INSERT request with context $h^{(k)}$:

$$M^{(k)} = \text{POSTINS}\big(M^{(k-1)}, \ \text{PREINS}(h^{(k)})\big), \quad (1)$$

where PREINS normalizes the raw context for insertion, and POSTINS updates the memory state according to maintenance policies.

**Retrieval Pipeline.** For a RETRIEVE request with query $q$, the system accesses the current memory state $M^{(k^*)}$ (where $k^*$ denotes the state after the most recent insertion):

$$c = \text{POSTRET}\big(M^{(k^*)}, \ \text{PRERET}(q)\big), \quad (2)$$

where PRERET formulates the retrieval signal from the user query, and POSTRET synthesizes the final context by retrieving and refining evidence to construct the final context $c$.

These four operators, anchored by the underlying structure of $M$, constitute the functional backbone of any *External Memory Module*. To isolate the impact of specific architectural decisions, we map these mathematical components to five factorized design dimensions (D1–D5) in the following taxonomy.

### 3.2. Design Aspects of External Memory Module

Implementation decisions for the operators in Eq. (1)–(2) jointly determine system performance. We identify five factorized dimensions to decompose this design space (see Appendix A.3 for more details):

***Memory Data Structure* (D1, $M$):** This dimension functions as the primary storage substrate, organizing incoming information into queryable units. The fundamental design choice lies in *Topology*: *Partitional* architectures store memories as discrete, independent chunks (e.g., flat vector stores), prioritizing fast similarity search; whereas *Hierarchical* architectures model explicit dependencies between entities (e.g., knowledge graphs), enabling multi-hop reasoning. Within the *External Memory Module* lifecycle, this structure governs the overhead of state updates and the physical efficiency of subsequent retrievals.

***Normalization Strategy* (D2, PREINS) and *Consolidation Policy* (D3, POSTINS):** In conjunction with D1, these dimensions realize the *Insertion Pipeline*, defining how memory state evolves over time. Specifically, D2 performs preprocessing to translate unstructured history into the discrete storable units. This stage determines the granularity and fidelity of the ingested information. Following insertion, D3 manages the *state evolution* of $M$ to ensure long-horizon stability and compactness. It implements maintenance policies such as Conflict Resolution, Decay Eviction for capacity

control, and Structural Refinement for hierarchical memories. Together, D2 and D3 define the computational cost and representational quality of the memory's insertion pipeline.

*Query Formulation Strategy* (D4, PRERET) and *Context Integration Mechanism* (D5, POSTRET): Operating upon the storage substrate D1, these dimensions operationalize the *Retrieval Pipeline*, governing how stored context is surfaced. Functionally, D4 acts as the alignment interface, bridging user intent with retrieval signals compatible with D1, such as generating embeddings for partitional stores or decomposing paths for hierarchical graphs. Upon retrieving evidence, D5 serves as the *output synthesizer* that executes the POSTRET operator, refining raw candidates into the final context via Reranking or Fusion. Ultimately, the synergy between D4 and D5 dictates the retrieval precision and the contextual coherence of the generated response.

By isolating these dimensions, *Neuromem* allows us to measure how computational cost is shifted between insertion and retrieval pipelines.

# 4. Methodology

We present our streaming evaluation framework, detailing the protocol and workload adaptation (§4.1), representative systems mapped to the taxonomy (§4.2), and the metrics and testbed (§4.3).

## 4.1. Streaming Protocol and Workloads

To model realistic deployment, the request stream $\mathcal{R}$ is instantiated as a strictly ordered time-series sequence. Unlike static benchmarks that decouple memory construction from usage, we enforce causality by sorting requests $r_i$ based on timestamps $\tau_i$. The system processes this stream sequentially: for an INSERT request, it invokes Eq. (1) to evolve the state from $M^{(k-1)}$ to $M^{(k)}$; for a RETRIEVE request, it executes Eq. (2) against the *current* state $M^{(k)}$. This interleaved execution ensures that a query $q_t$ is strictly limited to accessing evidence from contexts $\{h_{t'} \mid \tau_{t'} < \tau_t\}$ already integrated into the memory. By adhering to this temporal order, the protocol naturally prevents future leakage without requiring artificial visibility masks, while explicitly capturing the computational costs of state transitions.

We report results under a blocking online-consistency regime to make lifecycle attribution comparable across interchangeable operators. Asynchronous implementations can hide wall-clock maintenance latency, but they introduce freshness, visibility, and staleness trade-offs that require separate evaluation.

LOCOMO, LONGMEMEVAL, and MEMORYAGENT-BENCH are serialized into the stream $\mathcal{R}$ to preserve chronological ordering. Evaluation is triggered at fixed intervals (e.g., 20%) to capture performance evolution as memory accumulates, rather than only at the final state. Preprocessing details are in Appendix A.4.

## 4.2. System Instantiations

We reproduce representative memory modules by explicitly mapping their components to our D1–D5 taxonomy, covering TIM (Liu et al., 2023), MEMORYBANK (Zhong et al., 2024), MEMGPT (Packer et al., 2023), A-MEM (Xu et al., 2025), HIPPORAG (and HippoRAG 2) (Gutiérrez et al., 2024), MEMORYOS (Kang et al., 2025), LD-AGENT (Li et al., 2025), SCM (Wang et al., 2026), MEM0 and MEM0$^g$ (Chhikara et al., 2025), and SECOM (Pan et al., 2025). This decomposition transforms monolithic systems into atomic operators, allowing us to construct interchangeable variants and conduct granular ablations to isolate the impact of specific design decisions. New memory mechanisms can be integrated by implementing stage-compatible D1–D5 components, while collaborative multi-agent shared memory with concurrent synchronization and access-control semantics remains outside the current scope. Implementation details and abstraction boundaries are provided in Appendix A.2 and Appendix A.3.

## 4.3. Metrics and Platform

We use Token-level F1 with Porter stemming as the primary exact-fidelity metric and report granular Insertion Latency and Retrieval Latency for system efficiency. To separate lexical exactness from semantic adequacy, Appendix C.2.2 further provides an LLM-Judge analysis and task-type temporal breakdown; these complementary diagnostics help interpret, rather than replace, the controlled F1/latency comparisons. All experiments are executed on a unified serving stack hosting pangu-1b(Chen et al., 2025) on Huawei Ascend 910B NPUs and Llama-3.1-8B on Nvidia A6000 GPUs, using asynchronous scoring to avoid latency interference.

# 5. Experimental Analysis

This section evaluates *External Memory Modules* under the *streaming* protocol (Section 4.1) using *Neuromem*. The goal is attribution: we quantify how each lifecycle dimensions (D1–D5) influences answer quality and system cost as memory evolves, rather than treating implementations as black boxes.

*External Memory Modules* are highly sensitive to both operational hyperparameters (e.g., retrieval top-$k$) and workload characteristics (Packer et al., 2023). To enable deep attribution without combinatorial explosion, *Neuromem* focuses the full D1–D5 ablation on LOCOMO, while using LONG-MEMEVAL and MEMORYAGENTBENCH to cross-validate

structural trends (D1). In addition, Appendix C.1 provides targeted LONGMEMEVAL validation for insertion-side D2/D3 operators, which we treat as partial cross-workload support rather than full D2–D5 generalization evidence. We first distill our primary empirical observations in Section 5.1, followed by a granular analysis of each design dimension (D1–D5) via controlled ablations in Sections 5.2–5.6. Finally, we delineate the computational and theoretical boundaries of the current study in Section 5.7.

## 5.1. Findings

We distill the longitudinal evaluation results into five core empirical findings that govern the accuracy-latency trade-offs in streaming memory systems. Detailed decompositions supporting these conclusions are provided in subsequent sections.

**[F1] Temporal Degradation Persists Under Streaming Evaluation.** Performance tends to degrade as interaction history accumulates, establishing a baseline of noise that appears across the tested architectures. Computationally expensive consolidation policies driven by LLMs do not substantially mitigate this decay in our blocking online regime, exhibiting degradation rates comparable to unmaintained baselines. These proactive methods exhibit a similar degradation rate of approximately 22% as unmaintained baselines. This finding suggests that resolving semantic contradictions at ingestion time can become a premature optimization that incurs significant latency costs without clearly countering the degradation observed in long-horizon interaction.

**[F2] Storage Architecture Shapes Performance Bounds.** The underlying memory data structure is a dominant factor shaping the observed quality frontier. Multi-layer architectures that combine lexical and semantic indexing, typified by the `Inverted+Vector` design, perform strongly relative to single-signal baselines. These hybrid systems can justify their intrinsic latency by anchoring the efficiency frontier, whereas simpler approaches often struggle to capture necessary context. Our ablations suggest that downstream optimizations in query formulation or context integration have limited ability to compensate for information loss incurred at the storage level.

**[F3] Semantic Compression Can Be Lossy.** Aggressive abstraction during the normalization phase is often harmful under the tested setup. Transforming natural dialogue into rigid structured schemas causes substantial degradation, reducing F1 scores by over 50%. Our results indicate that structural schemas can discard subtle linguistic context and temporal markers essential for effective retrieval, making the preservation of raw textual texture more effective under these workloads.

**[F4] Generative Optimization Can Incur a Latency Tax.**

Incorporating generative steps into the retrieval pipeline imposes a substantial latency penalty with diminishing returns. Strategies such as query decomposition and multi-query expansion inflate processing times by an order of magnitude, often exceeding one second per turn. Despite this high computational cost, these methods yield negligible or even negative accuracy gains compared to direct processing. This inverse cost-benefit profile persists even when scaling to stronger backbones like Llama-3-8B, indicating current cost-inefficiency in generative retrieval enhancement under the blocking online regime.

**[F5] Heuristics Can Anchor the Efficiency Frontier.** In the constraint-heavy regime of online streaming, deterministic heuristics often offer better efficiency than generative interventions. Mechanisms like heat-based migration and heuristic context augmentation achieve parity or competitive reasoning accuracy compared to their model-based counterparts. Crucially, these heuristic methods operate with negligible latency overheads often measuring less than one millisecond. This suggests that a practical design strategy for streaming memory is to decouple state maintenance from expensive token generation processes when strict freshness and low latency are required.

## 5.2. Memory Data Structure

We analyze the impact of memory representations (D1) by instantiating the core data structures from the representative systems described in §4.2. These implementations span the Partitional and Hierarchical paradigms, extending from the Fifo Queue typified by SCM to the Linknote Graph underpinning A-MEM. To evaluate the intrinsic properties of these storage substrates, we integrate them as interchangeable modules within a unified serving pipeline, fixing all other dimensions (D2–D5) to minimal baselines. Figure 2 visualizes the resulting accuracy–latency landscape across the three workloads.

*Table 1.* **Temporal Degradation on LoCoMo.** Performance consistently declines across all structures as interaction history accumulates (R1→R5).

| Structure Type | Token-level F1 per Round | | | | | Degradation |
|---|---|---|---|---|---|---|
| | R1 | R2 | R3 | R4 | R5 | (△ R1→R5) |
| Fifo Queue | 0.169 | 0.128 | 0.118 | 0.109 | 0.094 | **-44.4%** |
| Queue+Segment | 0.395 | 0.362 | 0.356 | 0.349 | 0.338 | -14.4% |
| **Inverted+Vector** | **0.411** | **0.395** | **0.375** | **0.385** | **0.358** | **-12.9%** |

**O1: Performance degradation due to round accumulation is a consistent trend in our streaming workloads.** Our longitudinal analysis in Table 1 reveals consistent performance degradation as interaction history accumulates. From Round 1 to Round 5, even the robust `Inverted+Vector` structure suffers an approximate 13% decline as the F1 score falls from 0.411 to 0.358. In comparison, limited-capacity baselines like the

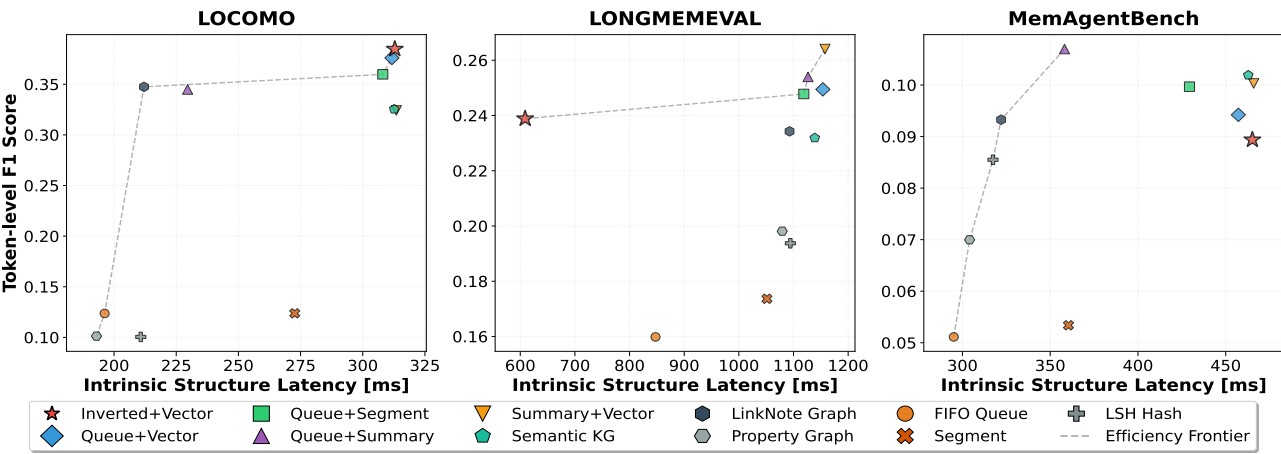

*Figure 2.* **Intrinsic Efficiency Frontier:** Plotting accuracy against **Intrinsic Structure Latency**. The **Inverted+Vector** anchors the Pareto frontier on reasoning benchmarks (Left, Center), whereas the trade-off inverts on maintenance-heavy tasks (Right) where lightweight queues achieve superior efficiency.

`Fifo queue` deteriorate by over 44% over the same period. This inverse correlation between history length and retrieval precision underscores that structural optimization alone does not fully arrest the noise accumulation observed in streaming, motivating active consolidation policies. A query-category breakdown further indicates that this degradation is driven mainly by distractor-sensitive queries rather than uniformly by all question types: standard queries drop from 0.6034 to 0.4736 (21.50%), and multi-answer queries drop from 0.3341 to 0.2797 (16.29%), whereas time-related queries start from a much lower baseline but change only from 0.1235 to 0.1195 (3.24%).

**O2: Multi-layer Partitional structures improve the *Intrinsic Return on Investment* (ROI).** Visualizing the accuracy–latency landscape (Figure 2) exposes distinct cost–benefit profiles. For reasoning benchmarks such as LOCOMO and LONGMEMEVAL, the Multi-layer `Inverted+Vector` design justifies its higher intrinsic latency. It anchors the efficiency frontier by converting this computational investment into resilience that can surpass Hierarchical graphs in recall while avoiding the substantial overhead associated with generative baselines. However, this return on investment inverts in high-churn environments like MEMORYAGENTBENCH. In these volatile streams, the maintenance overhead of vector indices becomes a liability, making the Single-layer `Queue+Summary` more favorable in both accuracy and speed. Consequently, Multi-layer designs are favorable for reasoning stability, whereas Single-layer approaches are attractive in scenarios requiring frequent maintenance.

### 5.3. Normalization Strategy

We anchor our evaluation on three distinct storage architectures: LSH Hash, Queue+Segment, and Property Graph. These configurations instantiate representative design paradigms from TIM, MEMORYOS, and MEM0$^g$, respectively, and serve as the default baselines for this and subsequent dimensions unless otherwise specified. We investigate three normalization strategies, selected to represent the most common preprocessing paradigms: `None` preserves the raw context; `Enrich` performs summarization; and `Rewrite` extracts triplets. Figure 3(a) illustrates the resulting cost-effectiveness evolution across rounds.

*Table 2.* **Impact of Normalization on LoCoMo (Rounds 1–5).** While `Rewrite` causes substantial accuracy degradation, `Enrich` incurs large insertion costs for transient or negligible gains. Best F1 scores are bolded.

| Backend | Strategy | Token-level F1 per Round | | | | | Latency |
|---|---|---|---|---|---|---|---|
| | | R1 | R2 | R3 | R4 | R5 | |
| Queue+Segment | None | **0.371** | **0.335** | **0.342** | 0.338 | 0.325 | **146 ms** |
| | Enrich | 0.369 | 0.343 | 0.334 | 0.344 | 0.324 | 580 ms |
| | Rewrite | 0.171 | 0.129 | 0.119 | 0.123 | 0.112 | 1552 ms |
| Property Graph | None | 0.347 | 0.353 | 0.318 | 0.311 | **0.301** | 1171 ms |
| | Enrich | **0.380** | 0.334 | 0.317 | 0.313 | 0.300 | 2567 ms |
| | Rewrite | 0.163 | 0.139 | 0.121 | 0.119 | 0.113 | 4491 ms |

**O3: Aggressive restructuring is often harmful under the tested setup.** Transforming dialogue into rigid triples using the `Rewrite` strategy causes substantial accuracy degradation. As detailed in Table 2, the F1 score for Queue+Segment drops from 0.371 with the `None` baseline to 0.171 using `Rewrite` in the first round, and this deficit widens by Round 5 where accuracy falls from 0.325 down to 0.112. Temporal reasoning suffers the most with performance plummeting to a near-random F1 level of approximately 0.033. This suggests that semantic compression can be lossy as structural schemas discard the linguistic context and temporal markers essential for vector retrieval. This failure incurs a substantial cost since `Rewrite` inflates the insertion latency for Queue+Segment by over 10× from 146 ms to 1552 ms. This inefficiency persists even when

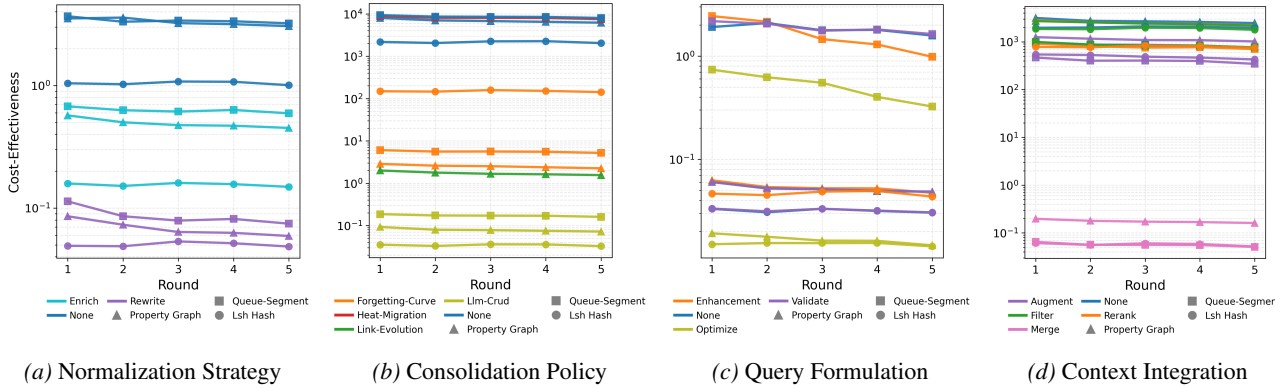

*(a)* Normalization Strategy      *(b)* Consolidation Policy      *(c)* Query Formulation      *(d)* Context Integration

*Figure 3.* **Breakdown of Cost-Effectiveness Evolution by Memory Lifecycle Dimension.** The plots illustrate the F1-per-second trajectory across five rounds for varying strategies in D2–D5.

scaling to stronger embedding backbones like E5-Large as discussed in Appendix C, indicating that preserving raw texture is more effective than structured abstraction in this setting.

***Robustness Check with*** **State-of-the-Art** *(SOTA)* **Mixture-of-Experts** *(MoE)* ***Model:*** We further validate our findings on a representative subset of the LoCoMo dataset using the Qwen-Plus model with the Property Graph architecture and a relaxed extraction limit of 10 triplets. As shown in Table 3, even with this enhanced setup, the `Rewrite` strategy reduces the mean F1 from 0.389 under the `None` baseline to 0.163. This performance degradation accompanies a latency increase from 111 ms to 1161 ms, suggesting that structural information loss remains a persistent bottleneck rather than only a symptom of insufficient model capacity.

*Table 3.* **Robustness Check with Qwen-Plus.** Even with a stronger backbone and relaxed extraction limits, `Rewrite` underperforms `None` while increasing insertion latency.

| Strategy | Max Triplets | Mean F1 | Latency |
|---|---|---|---|
| Pangu-`None` | N/A | 0.324 | 99 ms |
| Pangu-`Rewrite` | 5 | 0.132 | 1887 ms |
| Qwen-`None` | N/A | **0.389** | **111 ms** |
| Qwen-`Rewrite` | **10** | 0.163 | 1161 ms |

**O4: Enrichment is costly with limited returns.** The `Enrich` strategy imposes substantial costs for negligible gains, exemplified by Queue+Segment where insertion latency quadruples from 146 ms to 580 ms without improving F1 scores. Similarly, Property Graph observes only a transient benefit in the first round that disappears by Round 5, despite the summarization process doubling the write time to over 2.5 s. This negative trade-off is visualized in the cost-effectiveness trajectories of Figure 3(a), suggesting that the heavy computational tax of preprocessing drags down overall system efficiency. Since modern embeddings effectively capture semantics from raw segments, minimalism on the insertion pipeline remains a favorable strategy for long-horizon deployment.

### 5.4. Consolidation Policy

Consolidation updates the memory data structures state according to maintenance policies. We instantiate representative algorithms across three paradigms including `llm_crud` (Chhikara et al., 2025) for Conflict Resolution, `forgetting_curve` (Li et al., 2025) for Decay Eviction, and the `heat_migration` (Kang et al., 2025) and `link_evolution` (Chhikara et al., 2025) heuristics for Structure Enrichment. Table 4 contrasts these interventions against the unmaintained baseline.

*Table 4.* **Impact of Consolidation Policies.** We compare the baseline against proactive strategies using latency. The results show that while `CRUD` offers marginal accuracy improvements, it imposes a substantial post-processing latency penalty compared to the lightweight `Forgetting` curve.

| System | Strategy | Token-level F1 | | | Latency (ms) |
|---|---|---|---|---|---|
| | | R1 | R5 | Mean | |
| Property Graph | `None` | 0.348 | 0.269 | 0.300 | < 1 |
| | `CRUD` | **0.351** | **0.273** | 0.301 | 3,777 |
| | `Forgetting` | 0.343 | 0.270 | **0.303** | 120 |
| Queue+Segment | `None` | **0.370** | 0.317 | 0.341 | < 1 |
| | `CRUD` | **0.370** | **0.319** | **0.342** | 1,983 |
| | `Decay Eviction` | **0.370** | 0.318 | **0.342** | 61 |
| LSH Hash | `None` | 0.102 | 0.095 | 0.101 | < 1 |
| | `CRUD` | **0.103** | 0.095 | 0.100 | 2,894 |
| | `Forgetting` | 0.101 | **0.096** | **0.101** | < 1 |

**O5: In online streaming regimes, heuristic consolidation can be more efficient than generative maintenance.** The comparative analysis within the strict latency constraints of online streaming reveals that while generative strategies like `CRUD` aim to enhance consistency, they can become a substantial bottleneck for real-time pipelines. For the Property Graph architecture, enabling LLM-based resolution inflates the post-processing latency to 3,777 ms while only improving the mean F1 score by a marginal 0.001. Such multi-second overheads make active conflict resolution difficult for synchronous user interactions despite potential logical benefits. In contrast, lightweight heuristics achieve favorable efficiency-utility trade-offs suitable for online deployment. On the Queue+Segment system, the

`Decay Eviction` policy matches the peak accuracy of the expensive `CRUD` strategy at 0.342 but operates with a latency of only 61 ms compared to the 1,983 ms required for the generative approach. This suggests that algorithmic maintenance offers a more favorable strategy for online memory systems under our tested regime: it provides deterministic control for policy-driven forgetting without the large latency of generative models, effectively distinguishing intentional state management from the inevitable temporal degradation.

## 5.5. Query Formulation Strategy

The Query Formulation dimension serves as the translational interface that converts raw user intent into actionable retrieval signals. Beyond the standard `None` baseline, we examine three distinct paradigms: the `Validate` heuristic for query gating, `Keyword` extraction for optimization, and the `Decompose` method for generative enhancement. Table 5 reports the resulting streaming performance, specifically contrasting the query formulation overhead with architectural insertion costs.

*Table 5.* **Performance and Latency of Query Formulation Strategies.** Generative strategies incur substantial latency without yielding proportional accuracy gains over the baseline.

| System | Strategy | Token-level F1 | | | Latency (ms) | |
|---|---|---|---|---|---|---|
| | | R1 | R5 | Mean | PreRet. | Ins. |
| Property Graph | None | **0.342** | **0.273** | **0.300** | 100 | 5742 |
| | Keyword | 0.102 | 0.082 | 0.091 | 870 | 5394 |
| | Decompose | 0.320 | 0.246 | 0.279 | 1913 | 5182 |
| Queue+Segment | None | 0.371 | **0.318** | **0.341** | 161 | 187 |
| | Keyword | 0.114 | 0.083 | 0.099 | 1117 | 193 |
| | Decompose | 0.367 | 0.302 | 0.328 | 2209 | 209 |
| LSH Hash | None | 0.065 | 0.061 | 0.063 | 157 | 1963 |
| | Keyword | 0.030 | 0.030 | 0.031 | 1064 | 2056 |
| | Decompose | **0.093** | **0.090** | **0.094** | 2352 | 2016 |

**O6: Generative query formulation is costly under the tested setting.** The data in Table 5 indicates that deploying LLM-based strategies like `Keyword` extraction or `Decompose` often introduces efficiency overheads that can outweigh their benefits compared to direct processing. For instance, reducing queries to keywords leads to significant performance degradation in Queue+Segment, where the Mean F1 score drops from 0.341 using the `None` baseline to 0.099 with `Keyword`. This decline in accuracy is accompanied by a substantial latency increase, as the generation step adds between 870 ms and 1117 ms to the pipeline, contrasting sharply with standard vector lookups that take roughly 160 ms. Similarly, the `Decompose` strategy imposes substantial computational overhead while degrading performance. While Queue+Segment demonstrates inherent efficiency with insertion and retrieval latencies under 200 ms, enabling decomposition inflates the pre-retrieval formulation latency by over ten times to 2209 ms while resulting in a slightly lower F1 score of 0.328. This sug-

gests that for robust storage backbones, preserving semantic context via direct retrieval often offers a more favorable cost-benefit ratio than expensive generative interventions.

## 5.6. Context Integration Mechanism

Our analysis concludes with the Context Integration Mechanism, which acts as the interface bridging discrete retrieval results and the continuous generation process. We compare the lightweight heuristic `Augment` against the generative expansion strategy `Multi-query`, with detailed breakdowns provided in Table 6 and Table 7.

*Table 6.* **Impact of Context Integration.** Generative fusion incurs substantial latency overhead compared to heuristics. It increases context integration latency by orders of magnitude without delivering meaningful F1 gains.

| System | Strategy | Token-level F1 per Round | | | | | Mean F1 | Latency |
|---|---|---|---|---|---|---|---|---|
| | | R1 | R2 | R3 | R4 | R5 | | |
| Property Graph | None | **0.346** | 0.301 | **0.294** | 0.286 | 0.271 | **0.299** | < 1 |
| | Augment | 0.333 | **0.309** | 0.290 | **0.288** | 0.270 | 0.298 | < 1 |
| | Multi-query | 0.335 | 0.305 | 0.293 | 0.286 | **0.274** | 0.299 | 1700 |
| LSH Hash | None | 0.095 | **0.096** | **0.101** | **0.100** | **0.094** | **0.097** | < 1 |
| | Augment | 0.097 | 0.094 | 0.087 | 0.083 | 0.076 | 0.088 | < 1 |
| | Multi-query | **0.097** | 0.089 | 0.096 | 0.093 | 0.083 | 0.092 | 1584 |

**O7: Multi-query expansion incurs large latency overhead with diminishing returns.** While generative fusion is often hypothesized to improve robustness, our results in Table 6 show that it can become a large latency overhead under the tested backbone. For Property Graph, enabling `Multi-query` yields no improvement in Mean F1, stagnating at 0.299, yet it increases post-processing latency from negligible levels below 1 ms to approximately 1.7 s. The trend worsens for LSH Hash, where performance drops from 0.097 to 0.092 despite the same large latency penalty. This suggests that, at least for the default backbone, the cost is paid in seconds while the value is limited or negative.

*Table 7.* **Robustness Verification with Llama-3-8B.** Although `Multi-query` slightly outperforms `Augment`, it incurs a substantial latency cost compared to the heuristic baseline.

| Strategy | Token-level F1 per Round | | | | | Mean F1 | D5 Latency |
|---|---|---|---|---|---|---|---|
| | R1 | R2 | R3 | R4 | R5 | | |
| Augment | 0.339 | 0.301 | 0.289 | 0.283 | 0.268 | 0.296 | < 1 |
| Multi-query | **0.348** | **0.306** | **0.295** | **0.289** | **0.279** | **0.303** | 1327 |

***Robustness Check with Llama-3-8B:*** As detailed in Table 7, switching to the stronger Llama-3-8B backbone reveals a nuanced trade-off. While `Multi-query` successfully outperforms `Augment` by raising the Mean F1 from 0.296 to 0.303, this accuracy gain comes at a steep price. The integration latency surges from negligible levels ($< 1$ ms) to 1327 ms. This suggests that while stronger models can extract value from generative fusion, placing these steps on the retrieval critical path remains a costly strategy, likely suitable only when precision is paramount and latency is not a primary constraint.

## 5.7. Limitation

The scope of our analysis is primarily bounded by the computational intractability of iterative graph refinement where paradigms like HIPPORAG were excluded due to optimization latencies that scale poorly with memory size. Such overheads can extend execution times from minutes to hours effectively rendering these systems incompatible with the strict timeouts of real-time conversational pipelines. This exclusion reflects a mismatch with our evaluated online blocking regime rather than a negative judgment of their utility in offline or asynchronous settings. This constraint on algorithmic coverage is paralleled by an empirical limitation where the scarcity of datasets explicitly curated for dynamic memory evolution precludes the granular monitoring of intermediate states. Consequently our evaluation relies on end-to-end outcome metrics rather than precise state verification. Moreover, the absence of established definitions distinguishing short-term working memory from long-horizon storage prevents the application of hybrid consolidation strategies that could otherwise optimize memory state updates based on retention scope, identifying the formalization of these hierarchical tiers as a critical direction for future architectural research.

## 6. Conclusion

This paper investigates *External Memory Modules* in the realistic streaming regime where insertions interleave with retrievals. We introduce *Neuromem* to decompose the lifecycle into five dimensions, enabling granular attribution of answer quality and system latency. Our results reveal that the storage structure largely shapes the observed accuracy frontier, while content-destructive transformations often degrade performance despite their overhead. Crucially, we identify substantial latency overheads in generative optimizations under the tested blocking regime: while stronger backbones can extract utility from strategies like multi-query expansion, the computational cost remains high for real-time deployment compared to lightweight heuristics. Furthermore, we show that heuristic maintenance offers a favorable trade-off for online systems, providing deterministic control useful for future privacy-aware memory management without the latency of generative approaches. *Neuromem* establishes a reusable substrate for future work to optimize the full insert–maintain–retrieval–integrate loop under accumulated noise and evolving memory states in long-horizon interaction.

## Impact Statement

This work aims to advance the field of Machine Learning by focusing on the evaluation and optimization of External Memory Modules for Large Language Models under realistic streaming conditions.

From a societal and environmental perspective, our granular decomposition of the memory lifecycle identifies substantial efficiency costs in current generative designs. We demonstrate that aggressive semantic compression and post-retrieval fusion often incur large computational overheads for negligible or even negative performance gains. By establishing that lightweight structural heuristics can anchor the efficiency frontier, our findings offer a rigorous pathway toward more energy-efficient and scalable AI systems, directly supporting the principles of Green AI.

From an ethical and privacy perspective, the shift to a streaming protocol highlights the risks of unbounded data accumulation. Our analysis suggests that heuristic consolidation policies, such as decay eviction strategies, can provide more transparent control surfaces for future privacy-aware memory management while maintaining favorable latency profiles. However, we do not empirically validate privacy compliance or deletion guarantees in this work. We therefore frame these observations as design implications and encourage future research to evaluate privacy-preserving memory updates alongside accuracy and latency optimization.

## Acknowledgements

Our work is supported by Hubei Provincial Natural Science Foundation of China (No. 2026AFA002), NSFC-RGC under Grant 6246116033 and National Natural Science Foundation of China under Grant U25B2023.

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

# A. Supplement

This appendix provides additional background, reproducibility details, and extended experimental results that support the main paper.

## A.1. Related Work

### A.1.1. EXTERNAL MEMORY MODULES

We analyze twelve representative *External Memory Modules* along the five lifecycle stages (D1–D5) defined in Section 3. Consistent with our taxonomy, we categorize these systems into two architectural paradigms based on their topological organization: *Hierarchical and Graph-Structured Services* and *Partitional Memory Services*.

**Hierarchical and Graph-Structured Modules.** Systems such as HIPPORAG (and HippoRAG 2) (Gutiérrez et al., 2024), MEM0$^g$ (Chhikara et al., 2025), and A-MEM (Xu et al., 2025) align with the hierarchical paradigm by explicitly modeling the topological relationships between memory units. Rather than treating memories as independent vectors, these architectures construct semantic knowledge graphs (D1) where edges encode relational dependencies. This necessitates rigorous extraction-based normalization (D2) to transform raw text into triples or linked entities. Consequently, consolidation (D3) becomes a structural operation involving link evolution or graph densification, while integration (D5) leverages graph traversal algorithms to retrieve multi-hop context unreachable via simple similarity search.

**Partitional Memory Modules.** The remaining systems employ partitional architectures (D1), treating memory as a flat pool managed through composed indices (e.g., vector stores, temporal queues, keyword tables). Rather than modeling explicit topology, these services optimize system behavior through the strategic composition of retrieval structures. One prominent direction, adopted by MEMORYOS (Kang et al., 2025), MEMGPT (Packer et al., 2023), SCM (Wang et al., 2026), LD-AGENT (Li et al., 2025), and MEMORYBANK (Zhong et al., 2024), leverages this flexibility for *lifecycle and capacity management*. By organizing memory into temporal tiers (e.g., short-term queues vs. long-horizon storage), these systems implement dynamic consolidation policies (D3)—such as recursive summarization or operating-system-inspired paging—to handle long-horizon context streams within bounded resource constraints. Conversely, systems like MEM0 (Chhikara et al., 2025), TIM (Liu et al., 2023), and SECOM (Pan et al., 2025) utilize partitional indexing to enhance *semantic precision and consistency*. These approaches discretize inputs into independent atomic units (D2) and employ granular indexing strategies (D4) to support targeted updates. This design facilitates rigorous conflict resolution (D3), enabling the system to detect and correct contradictions via LLM-driven operations efficiently, a task often computationally prohibitive in dense graph structures.

### A.1.2. MEMORY BENCHMARK

We categorize existing benchmarks based on their evaluation protocols and measurement focus, contrasting them with our streaming approach. Table 8 summarizes these dimensions.

Most established benchmarks operate under an **offline protocol**, decoupling memory construction from querying. LOCOMO (Maharana et al., 2024) targets long-horizon information synthesis in dyadic dialogues, while LONG-MEMEVAL (Wu et al., 2025) focuses on recall consistency across multi-session chats. Both process static logs where memory is built in a single pass. MEMBENCH (Tan et al., 2025) similarly evaluates retrieval from a frozen memory pool. An exception is MEMORYAGENTBENCH (Hu et al., 2025), which supports incremental ingestion for agentic tasks (e.g., fact consolidation); however, it typically defers queries to the end of the session, failing to capture the complexity of fully interleaved production-retrieval cycles.

Existing benchmarks also vary significantly in evaluation granularity and measurement coverage. LOCOMO evaluates at the **model level** (end-to-end output), whereas others like MEMBENCH and LONGMEMEVAL focus on the **memory module's** isolation. Crucially, measurement often prioritizes accuracy over system cost. LOCOMO, LONGMEMEVAL, and MEMORYAGENTBENCH report only performance metrics (e.g., accuracy, consistency). While MEMBENCH includes simple efficiency metrics such as *Read Time* (RT) and *Write Time* (WT), it does not distinguish between model inference and memory maintenance costs, nor does it track efficiency evolution over time.

**Our Work** bridges these gaps by proposing a testbed at the **operator level** with a **streaming protocol**. By interleaving insertion and retrieval, we evaluate performance on evolving memory states $M^{(k_t)}$. Unlike prior works, we provide a dual-track assessment of both accuracy (F1) and efficiency through insertion/retrieval latency, explicitly decoupling memory

system overhead from the underlying LLM.

*Table 8.* Comparison of memory benchmarks across evaluation granularity, protocol, and measurement coverage

| Benchmark | Evaluation Granularity | Evaluation Protocol | Measurement Coverage |
|---|---|---|---|
| LoCoMo | Model level | Offline | Accuracy only |
| MemoryAgentBench | Module level | Incremental Insertion | Accuracy only |
| LongMemEval | Module level | Offline | Accuracy only |
| MemBench | Module level | Offline | Accuracy + Partial efficiency |
| **Our Work** | ***Operator level*** | ***Interleaved Insertion/Retrieval*** | ***Accuracy + Efficiency*** |

## A.2. Implementation Fidelity and Abstraction Boundary

Our goal is not to reproduce each system as a drop-in production service, but to instantiate its memory behavior as stage-compatible D1–D5 operators under the shared blocking protocol used in our benchmark. Table 9 summarizes the main preservation and abstraction choices. The table is intended to make the comparison boundary explicit rather than to claim full implementation equivalence.

*Table 9.* Implementation fidelity and abstraction boundary for representative memory systems

| Systems | Preserved in Neuromem | Abstracted or Simplified | Interpretation Boundary |
|---|---|---|---|
| TIM; MEM0/MEM0$^g$ | Structured fact extraction, vector or graph-backed retrieval, and conflict-aware memory update operators. | Product-specific serving wrappers, persistence layers, and non-essential orchestration logic are replaced by the shared D1–D5 runtime. | Results reflect the behavior of the memory operators under a common streaming protocol, not full service-level equivalence. |
| MEMORYBANK; MEMORYOS; MEMGPT | Temporal organization, summary or tier-based storage, and capacity-oriented memory migration or decay policies. | Interactive agent shells, user-interface logic, and long-running background maintenance are normalized into blocking stage calls. | Comparisons isolate lifecycle trade-offs while abstracting deployment-specific scheduling choices. |
| A-MEM; HIPPORAG/HippoRAG 2 | Graph or link-based memory structures, extraction-driven normalization, and graph-aware retrieval where compatible with the online budget. | Iterative global graph refinement and expensive multi-step maintenance are excluded from the main blocking benchmark when they exceed the evaluated latency regime. | Exclusion from the main online table indicates a regime mismatch, not evidence of ineffectiveness in offline or asynchronous settings. |
| LD-AGENT; SCM; SECOM | Keyword, summary, and controller-style operators are mapped to D2–D5 components for ingestion, retrieval, and context assembly. | Agent-specific planning loops and application-level policies are reduced to their memory-facing operators. | The benchmark compares memory-module behavior rather than complete agent policies. |

As a lightweight sanity check, we compared the Neuromem re-instantiated MEM0 module against a native/simple MEM0 pipeline on the same LoCoMo mini task. The three-segment F1 scores are close: 0.1902 for the Neuromem instantiation versus 0.1774 for the native/simple pipeline, with nearly identical retrieval latency (37.59 ms versus 36.98 ms). We use this result only as supporting evidence that the shared protocol preserves comparable observable behavior in this setting; it is not intended as a full equivalence claim for all MEM0 deployment features.

## A.3. Detailed Design Aspects of External Memory Modules

We deconstruct the twelve baseline systems evaluated in this study according to the five lifecycle stages (D1–D5). For each stage, we identify the primary design paradigms and categorize the baselines accordingly.

A.3.1. *Memory Data Structure* (D1)

We classify memory architectures into two dominant paradigms based on their *Memory Data Structure* (D1). `Partitional` services treat memory as a flat pool and rely on multiple complementary indices to support diverse access patterns. For instance, MEMORYOS combines FIFO queues with segmented storage to manage temporal tiers, while MEMORYBANK utilizes a combination of vector stores and summary buffers. This approach excels at predictable latency and scalability but may struggle with deep relational queries. In contrast, `Hierarchical` services embed memories into explicit relational structures. MEM0$^g$, for example, constructs semantic knowledge graphs where nodes represent entities and edges encode dependencies, enabling multi-hop retrieval beyond nearest-neighbor similarity. While offering richer reasoning primitives, these structures introduce additional maintenance complexity. Table 10 maps the evaluated systems to these structural paradigms.

*Table 10.* Summary of *Memory Data Structures* (D1) in representative models

| Structure Type | Representative Models | Specific Implementation |
| --- | --- | --- |
| Hierarchical/Graph | MEM0$^g$ | `semantic_inverted_knowledge_graph` |
| Partitional | MEM0
TIM
MEMORYOS
MEMORYBANK | `inverted_vectorstore_combination`
`lsh_hash`
`feature_queue_segment_combination`
`feature_summary_vectorstore_combination` |

A.3.2. *Normalization Strategy* (D2)

The *Normalization Strategy* (D2) transforms raw conversational inputs into storable units to balance *fidelity* with *indexability*. Strategies generally fall into three categories: `Direct Storage` (identity mapping), employed by models like MEMORYOS to maximize fidelity; `Enrichment`, which augments input via summarization or tagging to improve index compatibility, as seen in MEMORYBANK where raw interactions are summarized into events; and `Rewriting`, which converts unstructured dialogue into structured formats. For example, TIM and MEM0$^g$ extract relational triplets or atomic facts to support structured retrieval. While rewriting enhances relational reasoning, it increases dependence on upstream model quality compared to simpler enrichment or direct storage. Table 11 categorizes the baseline models by their specific normalization operators.

*Table 11.* Summary of *Normalization Strategies* (D2) in representative models

| Strategy | Representative Models | Specific Operator |
| --- | --- | --- |
| None/Embedding | MEMORYOS | `none` |
| Enrich | MEMORYBANK | `summarize` |
| Rewrite | TIM, MEM0$^g$ | `triplet_extract` |

A.3.3. *Consolidation Policy* (D3)

The *Consolidation Policy* (D3) manages the lifecycle of stored memories. We observe three main families: `Conflict Resolution` handles redundancy and contradictions; MEM0$^g$ implements this via LLM-driven CRUD operations to maintain consistency. `Decay Eviction` manages capacity constraints; MEMORYBANK uses an Ebbinghaus-inspired forgetting curve, while MEMORYOS employs heat-based migration to move data between tiers. `Structure Enrichment` evolves the relational topology, exemplified by A-MEM, which dynamically creates links between memory nodes. These policies improve long-horizon memory quality but often incur maintenance costs. Table 12 details the specific consolidation policies employed by each representative model.

*Table 12.* Summary of *Consolidation Policies* (D3) in representative models

| Policy | Representative Models | Specific Operator |
|---|---|---|
| None | SCM | `none` |
| Conflict Resolution | MEM0$^g$
TIM | `llm_crud`
`semantic_consolidation` |
| Decay Eviction | MEMORYBANK
MEMORYOS | `forgetting_curve`
`heat_migration` |
| Structure Enrichment | A-MEM | `link_evolution` |

### A.3.4. *Query Formulation Strategy* (D4)

The *Query Formulation Strategy* (D4) addresses the mismatch between user intentions and memory structures. Beyond `Direct Processing` (used by MEM0$^g$, MEMORYBANK, and TIM), systems employ more complex strategies. `Query Enhancement` and `Validation` can be used to decompose complex questions or verify retrieval necessity. `Query Optimization` is adopted by MEMORYOS, which extracts keywords from the query to target specific memory indices. These strategies transform the user's original query into a form better suited to the underlying memory structure, improving accuracy at the cost of latency. Table 13 lists the query formulation strategies adopted across the evaluated systems.

*Table 13.* Summary of *Query Formulation Strategies* (D4) in representative models

| Strategy | Representative Models | Specific Operator |
|---|---|---|
| None/Embedding | MEM0$^g$, MEMORYBANK, TIM | `embedding` |
| Validate | - | `validate` |
| Optimization | MEMORYOS | `keyword_extract` |
| Enhancement | - | `decompose` |

### A.3.5. *Context Integration Mechanism* (D5)

The *Context Integration Mechanism* (D5) determines what is presented to the generator model. Common strategies include `Filtering`, where systems like MEM0 use similarity thresholds to reduce noise; and `Reranking`, where MEMORYBANK uses time-weighted scoring to prioritize relevance. `Merging` strategies are critical in multi-tiered systems like MEMORYOS, which aggregates results from different storage layers. This stage manages the trade-off between *context quality* and *cost*: aggressive filtering reduces noise but risks harming recall, while complex reranking improves relevance at the expense of latency. Table 14 summarizes the context integration mechanisms implemented in the baseline models.

*Table 14.* Summary of *Context Integration Mechanisms* (D5) in representative models

| Strategy | Representative Models | Specific Operator |
|---|---|---|
| None | MEM0$^g$, TIM | `none` |
| Rerank | MEMORYBANK | `time_weighted` |
| Filter | MEM0 | `threshold` |
| Merge | MEMORYOS | `multi_tier` |

### A.4. Dataset Details and Preprocessing

We evaluate *External Memory Modules* on a suite of specialized datasets, adapting each to the *streaming* protocol to quantify the evolution of memory quality and cost over time.

**LoCoMo.** The LoCoMo dataset is structured around 10 core tasks, where each task comprises multiple sessions containing multi-turn dialogue sequences. Crucially, each task is associated with a specific set of tests accompanied by evidence labels that pinpoint the exact dialogue history required for reasoning. This hierarchical organization makes LoCoMo inherently

suitable for adaptation into a streaming format. In alignment with our streaming protocol, we serialize these multi-session dialogues into a continuous chronological stream, where evaluation queries are triggered immediately after their supporting evidence is ingested, ensuring strict temporal causality.

**LongMemEval.** To assess system stability over extended interaction periods, we utilize the `LongMemEvalOracle` subset of LONGMEMEVAL. The original benchmark structures each task as a sequence of $N$ historical turns followed by a single query. We adapt this for streaming evaluation by concatenating multiple independent tasks into a unified chronological stream. This approach creates a long-horizon timeline where questions are triggered at specific intervals, validating the system's ability to maintain and recall context across disjointed interaction sessions.

**MemoryAgentBench.** To evaluate the handling of evolving information, we incorporate the `Selective Forgetting` subset from MemoryAgentBench. Our analysis of the full benchmark revealed a significant presence of static factual knowledge; consequently, we exclusively selected the `Selective Forgetting` subset to construct our dataset, ensuring a focus on dynamic consistency. This subset simulates real-world scenarios where knowledge changes over time, requiring the memory module to correctly consolidate up-to-date facts from an incremental stream while effectively disregarding obsolete data.

To support our streaming evaluation, we parse the sequential fact updates into a discrete input stream and align the corresponding questions and ground-truth answers as synchronized evaluation checkpoints, ensuring that the memory system is tested on its real-time capacity to integrate new information and resolve contradictions.

### A.5. Algorithmic Details of Memory Operations

We provide pseudocode for the two atomic operations of *External Memory Module*, aligning with Equations (1) and (2).

#### A.5.1. MEMORY STATE UPDATE

Triggered by an INSERT request, this operation updates the memory state from $M^{(k-1)}$ to $M^{(k)}$ via preprocessing, insertion, and optimization.

---

**Algorithm 1** Algorithm of Memory Update Step (INSERT)

---

**Input:** Current memory state $M^{(k-1)}$, Historical context $h^{(k)}$
**Output:** Updated memory state $M^{(k)}$
**Hyperparameters:** Preprocessing config $\phi$, Optimization threshold $\theta$
 1: Step 1: Context Preprocessing
 2: $\quad \tilde{h} \leftarrow \text{PREINS}(h^{(k)}; \phi)$
 3: Step 2: Tentative Insertion
 4: $\quad M_{\text{inter}} \leftarrow \text{INSERT}(M^{(k-1)}, \tilde{h})$
 5: Step 3: Global Optimization (e.g., pruning, merging)
 6: $\quad M^{(k)} \leftarrow \text{OPTIMIZE}(M_{\text{inter}}; \theta)$
 7: **return** $M^{(k)}$

---

#### A.5.2. CONTEXT RETRIEVAL

Given a query $q$ at time $\tau$, retrieval accesses the latest memory state $M^{(k^*)}$ (where $k^*$ is the number of INSERT requests with $\tau_i \leq \tau$) and returns context $c$.

## B. Implementation Details

This appendix provides a comprehensive technical breakdown of the NEUROMEM framework to ensure the reproducibility and clarity of our experimental protocol. We first detail the engineering implementation of the interleaved insertion-and-retrieval protocol, focusing on the pipeline orchestration and backpressure mechanisms used to enforce temporal causality. Subsequently, we specify the hybrid hardware and software infrastructure utilized for different model backbones. Finally, we outline the standardized experimental workflow and the roadmap for open-source deployment.

---

**Algorithm 2** Context Retrieval Step (RETRIEVE)

---

**Input:** Current memory state $M^{(k^*)}$, Raw user query $q$
**Output:** Retrieved context $c$
**Hyperparameters:** Query refinement config $\psi$, Top-$k$ parameter $K$
 1: Step 1: Query Refinement
 2:   $\tilde{q} \leftarrow \text{PREQRY}(q; \psi)$
 3: Step 2: Similarity Search & Extraction
 4:   $\mathcal{C}_{\text{cand}} \leftarrow \text{SEARCH}(M^{(k^*)}, \tilde{q}, K)$
 5: Step 3: Context Aggregation
 6:   $c \leftarrow \text{REFINERETRIEVE}(\mathcal{C}_{\text{cand}})$
 7: **return** $c$

---

### B.1. Implementation of Interleaved Insertion-and-Retrieval Protocol

To support the streaming evaluation protocol defined in Section 4.1, we implemented a pipeline-based architecture in NEUROMEM that strictly enforces the causal ordering of the request stream $R$. As illustrated in Figure 4, the architecture comprises a central coordinator designated as the Main Pipeline, along with two functional sub-pipelines, namely the Insertion Pipeline and the Retrieval Pipeline, which jointly manage the full lifecycle of the External Memory Module.

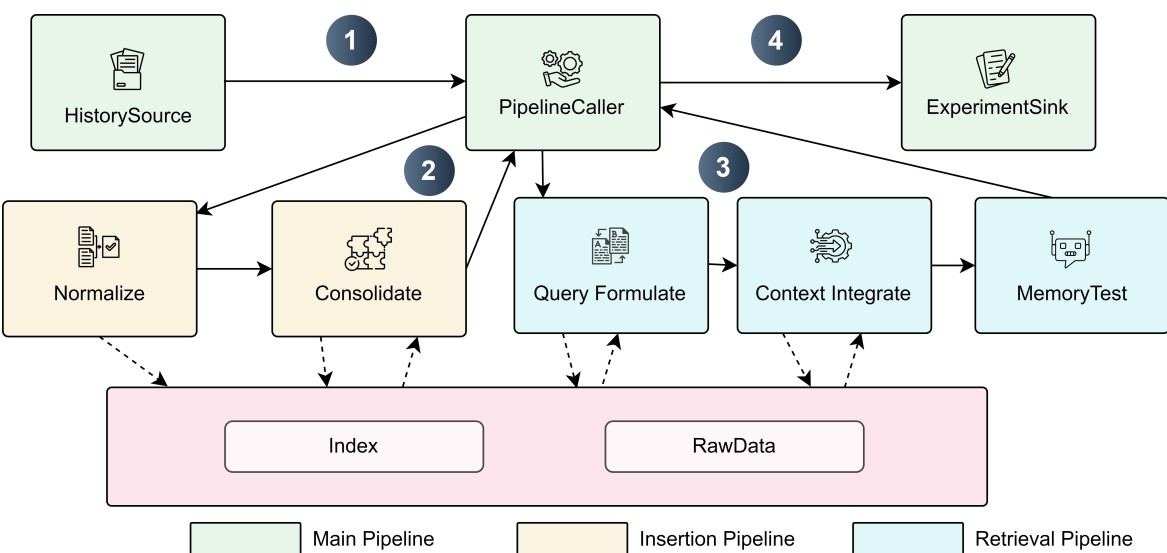

*Figure 4.* **Overview of the Neuromem Streaming Pipeline Architecture.** The central `PipelineCaller` holds the test dataset and orchestrates the interleaved execution. It employs a backpressure mechanism to block the historical data stream during memory maintenance or reasoning evaluation, ensuring strict temporal causality.

#### B.1.1. MAIN PIPELINE: THE ORCHESTRATOR

The Main Pipeline serves as the backbone of the testbed, governing the flow of time and data. It operates via a strict *backpressure* mechanism to ensure atomicity. It consists of three core components:

- **HistorySource:** Acting as the primary data source, this component is responsible for streaming interaction history data, such as context and facts, in a sequential manner. Unlike static batch loaders, it functions as a stream emitter. Driven by the backpressure mechanism from the downstream caller, the source pauses data emission whenever the processing pipeline is busy, which causes historical data to accumulate in a queue-based buffer rather than overflowing the system.

- **PipelineCaller:** Serving as the central coordinator, this component acts as the core scheduler to manage control flow and maintain the evaluation dataset comprising questions and ground truth answers. Rather than passively forwarding data, it actively monitors the incoming stream from the `HistorySource`. The logic follows a strictly blocking execution cycle that verifies specific state thresholds before triggering downstream processes.

1. **Insertion Trigger:** Upon receiving each incoming history item $h_t$, the coordinator blocks the stream and synchronously triggers the *Insertion Pipeline* to update the memory state.

2. **Threshold Check & Backpressure:** Following each insertion, the system verifies whether the current state satisfies preset evaluation checkpoints, such as reaching a segment boundary in LOCOMO or achieving a specific fact update count in the MEMORYAGENTBENCH dataset.

3. **Incremental Testing:** If a threshold is met, the `PipelineCaller` maintains backpressure on the `HistorySource` by keeping the historical data stream blocked while it initiates the *Retrieval Pipeline* to perform an incremental test. The stream remains blocked until the testing concludes, at which point the system proceeds to process the next historical item $h_{t+1}$.

- **ExperimentSink:** Functioning as the primary result collector, this component aggregates evaluation metrics, including Token-level F1 scores and latency statistics, from the testing process and serializes them for subsequent analysis.

### B.1.2. INSERTION PIPELINE: STATE EVOLUTION (D1–D3)

When activated by the `PipelineCaller` for a specific history item, this pipeline instantiates Eq. (1), $M^{(k)} =$ POSTINS$(M^{(k-1)}, \text{PREINS}(h^{(k)}))$, into three consecutive stages. The mainstream is blocked until these stages complete:

1. **Normalize:** Implementing the Normalization Strategy, this stage standardizes the raw context using the `PreInsert` operator. Depending on the configuration, the process ranges from simple text cleaning to complex semantic compression, such as extracting triplets via `extract.triple` or generating summaries using `transform.summarize`.

2. **State Update:** This stage executes the write operation defined by the Memory Data Structure. The processed data units are committed to the underlying storage substrates, such as an Index or RawData container, by invoking specific service interfaces like `fifo_queue.insert()` to finalize physical storage.

3. **Consolidate:** Corresponding to the Consolidation Policy, this stage maintains the long-horizon stability of the memory state following the write operation. Key maintenance tasks include resolving conflicts between new and old knowledge via `conflict_resolution.llm_crud` or removing obsolete information by applying policies such as `decay_eviction.forgetting_curve`.

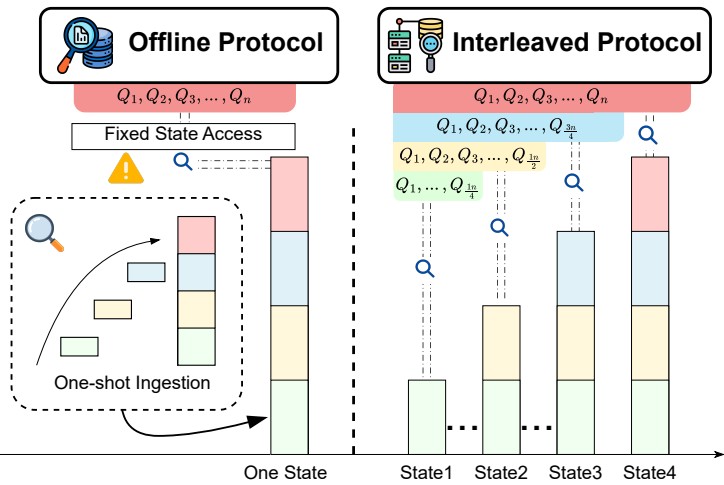

*Figure 5.* **Comparison of Evaluation Protocols.** Unlike the traditional **Offline Protocol** which queries a fixed memory state constructed in a single batch, our **Interleaved Protocol** performs retrieval at evolving memory states ($State1, State2, \ldots$), enforcing strict temporal causality.

B.1.3. Retrieval Pipeline: Reasoning and Evaluation (D1, D4–D5)

When the `PipelineCaller` determines that a test checkpoint is reached, it effectively freezes the memory state by blocking new insertions and initiates the Retrieval Pipeline. This mechanism implements the **Interleaved Protocol** illustrated in Figure 5 (Right), enabling assessments across dynamically evolving memory states rather than relying on a single fixed snapshot. The process instantiates Eq. (2), $c = \text{POSTRET}(M^{(k^*)}, \text{PRERET}(q))$, through the following sequential stages:

1. **Query Formulate:** Implementing the Query Formulation Strategy, this stage utilizes the `PreRetrieval` operator to convert the user query held by the Caller into executable retrieval signals. Common operations aim to bridge semantic gaps through methods such as `keyword_extract` or query rewriting via `optimize.rewrite`.

2. **Context Retrieval:** This stage executes the lookup operation defined by the Memory Data Structure. Using the transformed query signals, the system fetches candidate evidence from the underlying index or raw data storage via specific interfaces like `fifo_queue.retrieve()`.

3. **Context Integrate:** Corresponding to the Context Integration Mechanism, this stage employs the `PostRetrieval` operator to refine the fetched candidates. The process synthesizes the final context $c$ for generation by applying filtering or ranking mechanisms, such as `rerank.time_weighted` or `filter.threshold`.

4. **Memory Evaluation:** In the final step, the synthesized context is fed into the LLM to generate a response. The system immediately computes evaluation metrics, such as Token-level F1 scores, by comparing the prediction against the ground truth, and transmits the results back to the Main Pipeline.

By employing this strictly blocking Interleaved Protocol, *Neuromem* ensures that retrieval operations occur on a stable snapshot of the memory state $M^{(k)}$ without interference from incoming data, accurately simulating the real-time constraints of streaming applications.

## B.2. Hardware and Software Infrastructure

All experiments were conducted on high-performance computing platforms optimized for large-scale deep learning tasks. To support different model architectures and robustness checks, we utilized a hybrid hardware setup tailored to specific backbones:

- **Hardware Environment:**
  - **Huawei Ascend 910B NPU:** Dedicated to serving the `pangu_embedded_1b` model.
  - **Nvidia RTX A6000 GPU:** Dedicated to serving the `Meta-Llama-3.1-8B-Instruct` model.
  - **Storage:** 50 GB+ available disk space for dataset management and model weight storage.

- **Software Environment:**
  - **Operating System:** Linux.
  - **Programming Language:** Python 3.11 or higher.
  - **Core Models:**
    1. **Embedding Models:**
       · `BAAI/bge-m3`: Utilized as the default model for high-dimensional semantic representation.
       · `intfloat/e5-large-v2`: Utilized for robustness verification experiments.
    2. *Large Language Models* (LLMs):
       · `pangu_embedded_1b`: The primary backbone generative model, executed on the Ascend 910B platform.
       · `Meta-Llama-3.1-8B-Instruct`: Employed for cross-model validation and robustness checks, executed on the Nvidia A6000 platform.

### B.3. Implementation and Workflow Details

The source code and benchmark resources are available at `https://github.com/intellistream/neuromem-bench`. The framework is designed to facilitate direct streaming evaluation, allowing third-party researchers to decompose their custom memory modules and deploy them onto our evaluation pipeline for granular assessment. The standardized workflow is structured as follows:

1. **Environment Setup:** The framework requires a Conda environment with Python 3.11. The repository provides the installation scripts and dependency specifications:
   ```
   conda create -n mem python=3.11
   ```
   `https://github.com/intellistream/neuromem-bench`

2. **Dataset Acquisition:** We evaluate on `locomo`, `LongMemEvalOracle`, and the `Selective Forgetting` subset of MemoryAgentBench. The framework includes utilities to download and format benchmarks for streaming protocols; for example:
   ```
   neuromem-data download locomo
   ```

3. **Configuration:** System behaviors are fully configurable. Users can define model specifications and pipeline parameters (e.g., `benchmarks/experiment/config`) to customize the lifecycle dimensions (D1–D5).

4. **Execution & Analysis:** The core logic orchestrates the interleaved insertion and retrieval process. The suite separates execution (`benchmarks/experiment/script`) from analysis (`benchmarks/evaluation/scripts`), ensuring that metrics are computed consistently across different system instantiations.

## C. Detailed Experimental Results

This appendix provides additional analyses that complement the main experimental findings and clarify the robustness of the observed trends.

### C.1. Extended Analysis

We first examine the scalability behavior of graph-based memory maintenance to explain the latency patterns observed in the main text.

#### C.1.1. SCALABILITY ANALYSIS OF ITERATIVE GRAPH MEMORY

While graph-based architectures like HippoRAG demonstrate strong multi-hop reasoning capabilities, our analysis reveals significant scalability bottlenecks when applied to streaming protocols. Streaming deployment requires continuous graph maintenance, exposing the computational cost of iterative topology updates. Our profiling decomposes the insertion latency into two primary components: *Information Extraction Overhead* and *Synonymy Edge Construction*.

**Information Extraction Overhead.** Before graph construction, each incoming memory segment undergoes OpenIE processing, involving *Named Entity Recognition* (NER) and Triple Extraction. Since a single memory segment typically spawns multiple entities and facts, this step incurs a substantial fixed latency. Even with optimized local LLMs, the sequential execution of these generative calls imposes a base overhead of approximately 5-10 seconds per insertion, regardless of the memory size. This high entry cost makes sub-second real-time interaction challenging even at the start of the lifecycle.

**Synonymy Edge Construction (Quadratic Growth $O(N^2)$).** As the graph grows, the maintenance of semantic consistency becomes the dominant bottleneck. Identifying synonymy edges relies on a global *K-Nearest Neighbors* (KNN) search. Critically, the number of entities $N$ grows significantly faster than the number of dialogue turns due to the multiplicative factor of extraction. At each insertion step $t$, the system computes pairwise similarity scores for $N_t$ accumulated entities, resulting in $O(N_t^2 \cdot d)$ complexity. As illustrated in Figure 6, while the Information Extraction cost remains constant, the Synonymy Construction cost exhibits quadratic growth. For a history of 5,000 dialogues, the insertion latency can rise to over 100 seconds, making the system unsuitable for the evaluated blocking online setting.

**Contextual Integration ($O(I \cdot |E|)$).** Following graph updates, retrieval executes the *Personalized PageRank* (PPR) algorithm. While efficient for sparse graphs, the iterative convergence still imposes a latency floor of approximately 0.5 to 1.0 seconds for mid-sized graphs ($N \approx 25,000$), further straining the total response time.

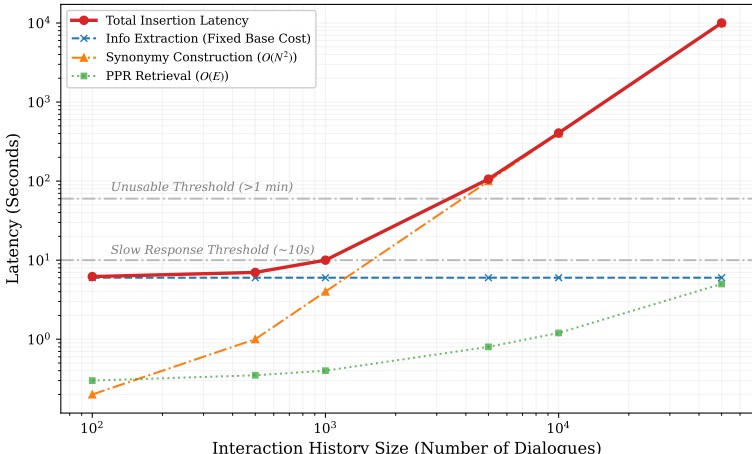

*Figure 6.* **Scalability Analysis of Iterative Graph Memory.** The log-log plot decomposes insertion latency into fixed and dynamic components. The *Information Extraction* imposes a high initial latency floor due to LLM calls. As memory scales, the *Synonymy Edge Construction* dominates with quadratic growth, causing total latency to exceed 1 minute beyond 5,000 dialogues. The *PPR Traversal* remains relatively efficient but adds to the retrieval latency.

In conclusion, iterative graph memory faces a two-fold scalability barrier under the evaluated online protocol: a high constant latency floor due to generative extraction and quadratic latency growth from non-incremental graph maintenance. Effective streaming deployment requires asynchronous extraction pipelines and incremental graph update algorithms (e.g., local neighborhood search) to decouple maintenance costs from total memory size.

## C.2. Robustness Experiment

We further evaluate whether the observed trends remain stable under stronger embedding and generation backbones and across targeted cross-workload validation.

### C.2.1. SUBSET CROSS-WORKLOAD VALIDATION ON LONGMEMEVAL

To stress-test whether the LoCoMo-centered observations are dataset artifacts, we conduct a targeted validation on LONG-MEMEVAL. We focus on D2 and D3 because these insertion-side operators expose the key accuracy–cost trade-offs most directly affected by workload content and memory growth. This experiment is intended as subset-scale cross-workload evidence, not as a complete D2–D5 sweep outside LoCoMo. Rather than presenting a new leaderboard-style table, we align the evidence with the lifecycle taxonomy used throughout the paper: *Normalization Strategy* (D2) introduces a clear `pre_insert_ms` latency overhead, while *Consolidation Policy* (D3) shows that LLM-driven maintenance is much more expensive than lightweight forgetting.

*Table 15.* D2 normalization latency overhead on LONGMEMEVAL

| Structure | None | Enrich | Rewrite | Enrich $\Delta$ | Rewrite $\Delta$ |
|---|---|---|---|---|---|
| Property Graph | 42.1 ms | 669.1 ms | 475.4 ms | +627.0 ms (+1490%) | +433.3 ms (+1029%) |
| Queue+Segment | 114.0 ms | 709.2 ms | 506.2 ms | +595.2 ms (+522%) | +392.2 ms (+344%) |
| LSH Hash | 77.0 ms | 670.9 ms | 487.3 ms | +593.9 ms (+771%) | +410.3 ms (+533%) |

For the Queue+Segment backend, this latency increase coincides with lower answer quality: F1 drops from 12.5% under `None` to 11.9% under `Enrich` and 8.3% under `Rewrite`. The additional cost is concentrated in the D2 preprocessing path: `Enrich` spends roughly 600 ms per round on summarization, while `Rewrite` spends roughly 400 ms per round on triplet extraction.

The D3 comparison shows a similar pattern. The `CRUD` policy spends about 439 ms in `post_insert_ms` for LLM-based conflict checking, whereas `Forgetting` adds almost no extra cost. Manual error inspection suggests two recurring failure

*Table 16.* D3 consolidation latency on Queue+Segment in LONGMEMEVAL

| Strategy | Total per Round | Δ vs. None | Overhead |
|---|---|---|---|
| None | 519.4 ms | – | – |
| Forgetting | 525.6 ms | +6.2 ms | +1.2% |
| CRUD | 962.9 ms | +443.5 ms | +85.4% |

modes behind the D2 quality drop. First, numeric facts can evaporate during compression: in one soccer example, `None` preserves "3 goals" and "2 assists" and answers 5 correctly, while `Rewrite` keeps only concept-level triples and `Enrich` shifts focus to unrelated preferences. Second, summarization can drift toward frequent but irrelevant topics: in a pet-care example, the raw dialogue preserves a \$50 vet visit and \$25 flea medication, whereas compressed memories either drop the amounts or mix them with an unrelated \$35 litter detail. Together with the three-workload D1 frontier in Figure 2, these LongMemEval results show that the main lifecycle conclusions are not driven solely by LoCoMo: the same accuracy–cost pattern reappears for insertion-side operators under a distinct long-memory workload. We therefore use this experiment as targeted cross-workload support for the bounded claims in the main paper, while keeping the full combinatorial sweep centered on LoCoMo for cost control.

### C.2.2. COMPLEMENTARY LLM-JUDGE ANALYSIS

We complement token-level F1 with an LLM Judge to separate exact lexical fidelity from semantic adequacy. The comparison reveals where F1 is intentionally strict and where semantic scoring provides additional diagnostic signal.

*Table 17.* Task-type sensitivity of Token-F1 and LLM Judge scores

| Task Type | Token-F1 | LLM Judge | Δ | Δ % |
|---|---|---|---|---|
| Standard | 0.3861 | 0.5321 | +0.1460 | +37.8% |
| Clean-Comments | 0.1453 | 0.2851 | +0.1398 | +96.2% |
| Multi-Answer | 0.2308 | 0.3489 | +0.1181 | +51.2% |
| Time-Related | 0.0953 | 0.0732 | -0.0222 | -23.3% |

Table 17 shows that Clean-Comments receives the largest relative correction from the judge, indicating that lexical F1 can under-credit cleaned or rewritten conversational evidence. Time-Related questions are the only category where the judge score is lower than F1, consistent with the fixed-form nature of date and numeric answers where semantic evaluation is stricter.

*Table 18.* False-negative attribution for low-F1 but semantically valid cases

| Error Category | Proportion | Count |
|---|---|---|
| Lexical Substitution | 72.5% | 145 |
| Format Variation | 11.5% | 23 |
| Partial Omission | 7.5% | 15 |
| Granularity Mismatch | 5.5% | 11 |
| Correct Elaboration | 3.0% | 6 |

Among 1,819 cases with F1 below 0.25 but Judge score at least 0.5, a 200-example attribution sample shows that 84% of the apparent failures come from lexical substitution or formatting variation rather than semantic errors (Table 18). This supports using F1 as a strict low-level probe while using the judge to identify semantically correct paraphrases.

The degradation curves in Table 19 further show that F1 decays faster than the judge score. For the Inverted+Vector configuration, the fitted slope is -0.01155 per round for F1 and -0.00849 per round for the judge, so exact lexical matching degrades about 1.36× faster than semantic adequacy. This reinforces the role of F1 as a deliberately strict system-level probe while explaining why the judge provides complementary evidence.

### C.2.3. ROBUSTNESS OF NORMALIZATION STRATEGY

To rule out the possibility that the poor performance of the `Rewrite` strategy was due to limited model capacity, we replicated the experiment on LoCoMo using the stronger `intfloat/e5-large-v2` embedding model.

*Table 19.* Relative degradation slopes under Token-F1 and LLM Judge

| Round | F1 Relative Drop | Judge Relative Drop | Ratio |
|-------|------------------|---------------------|-------|
| R2 | -12.14% | -7.47% | 1.63× |
| R3 | -16.21% | -11.72% | 1.38× |
| R4 | -17.77% | -12.07% | 1.47× |
| R5 | -23.81% | -16.88% | 1.41× |

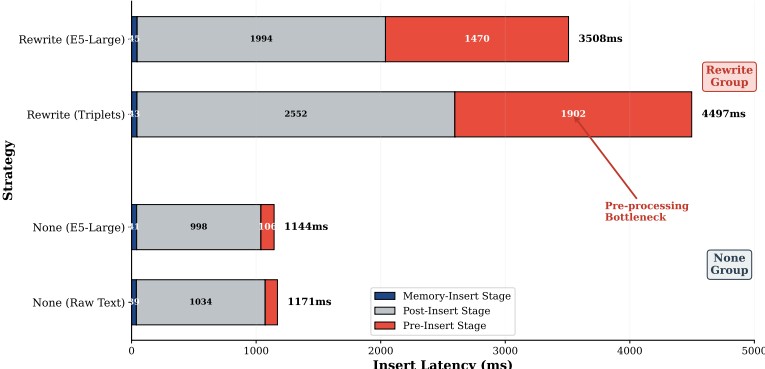

*Figure 7.* **Insert Latency Breakdown by Normalization Strategy.** The horizontal bars compare the time cost of ingestion stages. The `Rewrite` strategy incurs a large bottleneck, taking nearly 2 seconds just to extract triplets before any storage operation occurs. In contrast, the `None` baselines have negligible pre-processing costs.

*Table 20.* **Robustness Check on Normalization with E5-Large.** Even with a superior embedding backbone, the `Rewrite` strategy consistently underperforms the `None` baseline across all rounds. The latency breakdown reveals a heavy preprocessing cost for triplet extraction.

| Normalization | Token-level F1 per Round | | | | | Mean F1 | Insert Latency |
|---|---|---|---|---|---|---|---|
| | R1 | R2 | R3 | R4 | R5 | | |
| **None (Raw Text)** | **0.326** | **0.332** | **0.320** | **0.319** | **0.297** | **0.319** | **1,171 ms** |
| Rewrite (Triplets) | 0.168 | 0.128 | 0.120 | 0.116 | 0.111 | 0.129 | 4,497 ms |
| Δ *vs. Baseline* | -48.5% | -61.4% | -62.5% | -63.6% | -62.6% | -59.7% | +284% (Slower) |

The results, presented in Table 20 and visualized in Figure 7, confirm that the performance gap is tied to the structural representation rather than the embedding quality. The `Rewrite` strategy suffers a severe drop in F1 score (Mean $\Delta \approx -60\%$) compared to the raw text baseline.

Crucially, the latency breakdown in Figure 7 exposes the root cause of the inefficiency. While the core memory insertion time is negligible for all methods, the `Rewrite` strategy is dominated by the **Normalization Stage**, which demands approximately 1,902 ms to extract triplets from the raw text. This structural overhead results in a total insertion latency that is nearly 4× slower than the baseline (4,497 ms vs 1,171 ms). This reinforces our conclusion that for embedding-based retrieval, preserving the rich semantic texture of raw text is more effective than aggressive structural normalization in this setting, both in terms of accuracy and efficiency.

### C.2.4. ROBUSTNESS OF CONTEXT INTEGRATION STRATEGY

To investigate whether the high latency of generative fusion was merely an artifact of our default backbone's inference speed (pangu-1b), we replicated the Context Integration experiment using the faster and more capable **Llama-3-8B** model.

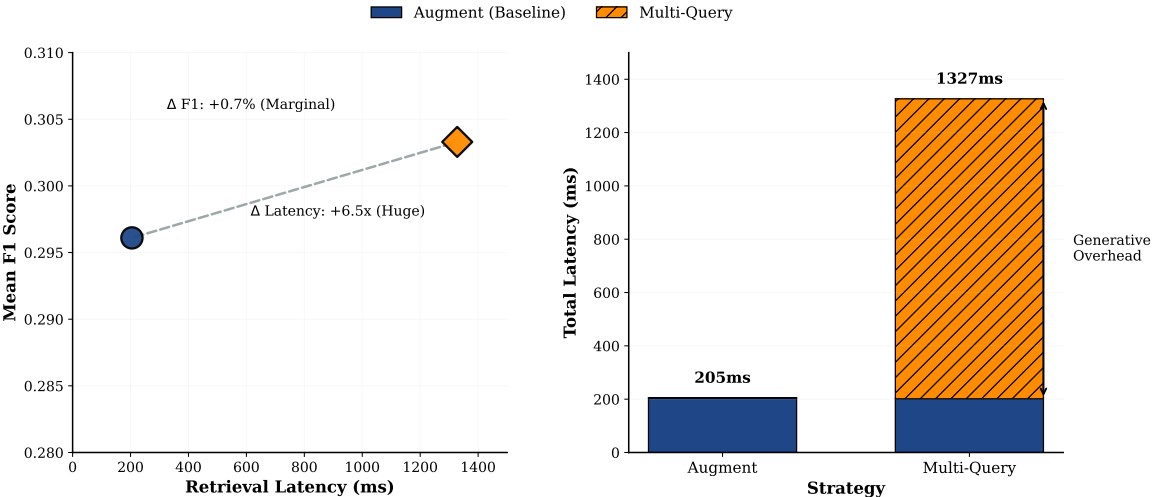

*Figure 8.* **Efficiency-Accuracy Trade-off with Llama-3-8B.** The left panel illustrates the disproportionate cost of generative fusion: a marginal mean F1 gain (+0.7%) requires a large 6.5× increase in latency. The right panel decomposes this latency, revealing that the cost is driven almost entirely by the Generative Overhead (1.1s) during the Context Integration stage, while the base retrieval time remains constant.

*Table 21.* **Robustness Check on Context Integration with Llama-3-8B.** Performance across all 5 rounds confirms that `Multi-Query` yields consistent but marginal gains. The latency breakdown highlights that the cost is driven entirely by the generative Context Integration stage, which adds over 1 second of overhead compared to the `Augment` baseline.

| Integration Strategy | Token-level F1 per Round | | | | | Mean F1 | Retrieval Latency Breakdown | |
|---|---|---|---|---|---|---|---|---|
| | R1 | R2 | R3 | R4 | R5 | | Context Integration | Total |
| `Augment` (Baseline) | 0.339 | 0.301 | 0.289 | 0.283 | 0.268 | 0.296 | ∼0.3 ms | **205 ms** |
| `Multi-Query` | **0.348** | **0.306** | **0.295** | **0.289** | **0.279** | **0.303** | 1,125 ms | 1,328 ms |
| *Impact (Δ vs Baseline)* | **+2.6%** | **+1.7%** | **+2.1%** | **+2.1%** | **+4.1%** | **+2.4%** | **Large Overhead** | **6.5× Slower** |

The results, visualized in Figure 8 and detailed in Table 21, indicate that the efficiency bottleneck is structural rather than model-dependent. Expanding the evaluation to the full five-round lifecycle reveals that while the `multi_query` strategy consistently outperforms the baseline—raising the Mean F1 score from 0.296 to 0.303—this advantage is slight. The strategy achieves a 4.1% relative improvement in the final round, but this accuracy benefit comes at a disproportionate cost. The total retrieval latency spikes from a manageable 205 milliseconds in the heuristic `augment` baseline to 1,328 milliseconds with generative fusion, representing a 6.5-fold increase in processing time.

Crucially, the granular breakdown identifies the root cause of this bottleneck. As shown in the right panel of Figure 8, while the baseline requires less than 1 millisecond for context integration, the generative approach demands approximately 1,125 milliseconds to synthesize and rerank the retrieved contexts. Consequently, the Context Integration stage alone accounts for the vast majority of the total latency. This finding reinforces our conclusion that generative integration introduces a substantial latency overhead: upgrading the model backbone reduces absolute inference time but does not eliminate the architectural overhead required for generative fusion.

