# OpenReview forum: "Neuromem: A Granular Decomposition of the Streaming Lifecycle in External Memory for LLMs"
_ICML.cc/2026/Conference — ICML 2026 regular_

### Official Review · Reviewer_pboU · 2026-03-11

**Soundness:** 3
**Presentation:** 4
**Significance:** 3
**Originality:** 3
**Overall Recommendation:** 5
**Confidence:** 3

**Summary:**

This paper introduces Neuromem, a testbed and evaluation framework for studying external memory modules for LLMs under a streaming, interleaved insertion-and-retrieval setting rather than the more common static evaluation setup. The paper’s main contribution is a decomposition of the memory lifecycle into five design dimensions (data structure, normalization, consolidation, query formulation, and context integration) and an empirical study of interchangeable variants within a shared serving stack on Locomo, Longmemeval, and Memoryagentbench. The central empirical message is that performance generally degrades as memory grows over time; storage architecture sets the main quality ceiling; aggressive normalization and generative query/context-integration steps often impose large latency costs for little or negative accuracy benefit; and lightweight heuristics can dominate more expensive LLM-based maintenance strategies in online settings.

**Compliance With Llm Reviewing Policy:**

Affirmed.

**Final Justification:**

After considering both the paper and the rebuttal, I am increasing my recommendation to 5 (accept). The paper’s main strengths are its original lifecycle-aware framing of streaming external memory, its practical and well-executed empirical design study, and its clear presentation; my main concern had been that some conclusions were stated too broadly for the empirical scope, but the rebuttal addressed that concern adequately by narrowing the claims, clarifying the intended scope, and adding supporting evidence.

Overall, the rebuttal increased my confidence in the paper’s soundness and significance rather than changing my view of its core contribution. I still see this primarily as a strong benchmarking and design-study paper, but with the revised framing, I think it makes a useful and credible contribution that others are likely to build on.

**Key Questions For Authors:**

1. The strongest claims in the paper concern broad design conclusions, such as the superiority of certain storage architectures and the inefficiency of some generative memory operations. Since the full D1–D5 study is conducted primarily on LOCOMO, how confident are you that these conclusions generalize across substantially different task settings rather than reflecting properties of that benchmark? A convincing answer here would increase my confidence in the breadth of the paper’s empirical claims.

2. Several of the paper’s conclusions are framed quite categorically (for example, around dominant architectures or fundamentally inefficient components). Could you clarify which of these should be interpreted as claims about your specific streaming testbed and latency regime, versus claims you believe hold more generally across external-memory systems? A more carefully scoped answer would help calibrate the paper’s significance and soundness.

3. The paper argues that lightweight heuristics often outperform more expensive LLM-based maintenance strategies in online settings. To what extent do you think this is a contingent result of the current cost-quality tradeoff in your implementation, versus a more durable conclusion about streaming memory systems? If you can show that the negative result is robust across implementation choices, that would strengthen the paper considerably.

4. Because the paper is primarily a benchmarking and design-study contribution, its value depends heavily on whether the framework will be useful beyond the exact methods instantiated here. How easily can Neuromem accommodate newer or qualitatively different memory mechanisms, and are there important classes of external-memory systems that your decomposition does not currently capture well? A strong answer would increase my assessment of the framework’s long-term usefulness.

5. One of the most practically relevant findings is that memory quality degrades as stored information accumulates over time. Can you better characterize the main source of this degradation (e.g., retrieval failure, noisy normalization, consolidation errors, or context integration bottlenecks) and whether you believe it reflects a benchmark artifact or a more general systems bottleneck? This would help clarify whether the paper’s main contribution is diagnostic insight into a general problem or primarily an empirical observation about the tested stack.

**Limitations:**

No. The paper discusses computational and theoretical boundaries of the study, but it would be stronger if it more explicitly acknowledged limits to generalization beyond the chosen serving stack, architectures, and benchmarks, and briefly discussed possible downstream risks of making long-horizon memory systems easier to optimize and deploy.

**Strengths And Weaknesses:**

**Strengths.** The paper tackles a useful and underexplored evaluation problem: most prior work treats external memory in a static regime, whereas real deployments are streaming and stateful, so the proposed lifecycle-aware framing is well motivated and potentially valuable to the community. A major strength is the paper’s decomposition of external memory systems into modular design dimensions, which makes the evaluation more diagnostic than leaderboard-style comparisons of monolithic systems. The empirical study is also reasonably rich and leads to actionable conclusions, for example that multi-layer inverted+vector structures anchor the quality frontier on reasoning-heavy benchmarks, while generative normalization, query decomposition, and multi-query integration often act as latency traps with weak returns. The paper is generally clear, and the attempt to measure both token-level F1 and insertion/retrieval latency within a unified serving stack makes the contribution practically relevant rather than purely taxonomic.

**Weaknesses**. The main weakness is that the paper is stronger as an empirical benchmarking and design-study paper than as a validation of broad scientific claims about external memory in general. Many of the strongest conclusions (for example that semantic compression is reliably destructive or that generative optimization is fundamentally inefficient) are drawn from a specific serving stack, a limited set of architectures, and a benchmark suite that, while diverse, still covers only a small slice of realistic deployment settings. The study also deliberately restricts the full D1–D5 ablation to LOCOMO and uses the other datasets mainly to cross-validate structural trends, which limits how confidently some of the broader conclusions can be generalized across task settings. In addition, several claims are stated quite strongly in the writing (“strictly superior,” “strictly dominant,” “fundamental inefficiency”) relative to the empirical scope of the evidence. In particular, while the reported results are convincing within the proposed streaming testbed, the current empirical scope does not seem broad enough to justify claims that certain design choices are universally superior or fundamentally inefficient across external-memory systems more generally.

---

> ### Author Rebuttal · Authors · 2026-03-31
>
> Thank you for the thoughtful feedback. We agree the paper is best positioned as an evaluation and design-study contribution, and we will revise the manuscript accordingly.
>
> ## Response to Weaknesses
>
> > W1. Claims are too broad relative to empirical scope
>
> We agree and will narrow claim scope throughout the paper. We will replace categorical wording with condition-aware wording tied to our tested setting (streaming, blocking, online consistency constraints, and current serving stacks). We will explicitly position our conclusions as empirical findings under these conditions, not universal laws across external-memory systems.
>
> > W2. Full D1-D5 ablation is concentrated on LoCoMo
>
> You are right. We used LoCoMo for full combinatorial ablations due to cost, then used additional datasets for trend cross-checks. To strengthen breadth, we ran additional subset validation on LongMemEval and observed directionally consistent trends for both structure ranking and quality-latency trade-offs.
>
> | Evaluation Focus | Configuration | F1 Score | Execution Latency |
> |---|---|---|---|
> | Storage Structure | Queue Segment Base | 12.5% | 114ms |
> | Storage Structure | Property Graph Base | 5.2% | 42ms |
> | Storage Structure | LSH Hash Base | 5.7% | 77ms |
> | Generative Operator | Plus CRUD Consolidation | 7.1% | 962ms |
>
> We will add these results, report subset size/protocol and label them as partial validation.
>
> > W3. Overly strong wording ("strictly superior", "fundamental inefficiency")
>
> Agreed. We will revise wording to avoid overclaiming. For example:
> - "strictly superior" -> "consistently stronger under tested conditions"
> - "fundamental inefficiency" -> "currently cost-inefficient in blocking online regimes"
>
> > W4. Current scope does not support universal superiority claims
>
> Agreed. We will explicitly separate what is likely transferable from what is regime-specific, and add a clearer limitations paragraph on generalization boundaries.
>
> ## Response to Questions
>
> > Q1. How confident are we about generalization beyond LoCoMo?
>
> Our confidence is moderate and evidence-based rather than absolute. We do not claim universal transfer. Instead, we claim directionally consistent behavior under similar online constraints. In LongMemEval subset validation, queue-segment structures remained stronger than graph/hash alternatives, while expensive generative consolidation increased latency without corresponding quality gains.
>
> > Q2. Which conclusions are testbed-specific vs more general?
> We will explicitly classify conclusions into two groups in the revision.
>
> 1. Findings likely to transfer across many memory systems:
> - Structural choices in D1 strongly shape the downstream quality ceiling.
> - Lossy compression can discard useful lexical detail and hurt exactness.
>
> 2. Findings specific to our streaming, blocking regime:
> - Generative maintenance often incurs significant latency overhead.
> - Lightweight heuristics can be preferable when strict freshness and low latency are required.
>
> > Q3. Is the latency-tax conclusion durable or implementation-contingent?
>
> Our claim is practical, not universal. Under current serving stacks and online blocking constraints, generative maintenance showed a large latency premium, about 10x with Qwen Plus in our setup. We agree this trade-off can shift with future model/system improvements and asynchronous deployment regimes, i.e., asynchronous read and write operations.
>
> > Q4. Extensibility of Neuromem and unsupported memory classes
>
> Neuromem is designed for extensibility through modular D1-D5 operators and a plugin registry. New mechanisms can be integrated by implementing stage-compatible components without changing the core evaluator. We will also explicitly state currently under-covered classes, including collaborative multi-agent shared memory with concurrent synchronization and access-control semantics.
>
> > Q5. Main source of temporal degradation
>
> Our diagnostics indicate retrieval under distractor density is the dominant contributor in our tested stack. We verified this by disaggregating query categories across rounds.
>
> | Category | R1 | R5 | Relative Change |
> |---|---|---|---|
> | Standard | 0.6034 | 0.4736 | 21.50% drop |
> | Multi-Answer | 0.3341 | 0.2797 | 16.29% drop |
> | Time-Related | 0.1235 | 0.1195 | 3.24% drop |
>
> The strongest degradation appears in semantically distractor-sensitive queries, while time-related queries are more stable.
>
> ## Response to Limitations Comment
> We appreciate this suggestion and will expand limitations in two ways.
>
> 1. Generalization boundaries:
> - We will explicitly state that conclusions are tied to selected stacks, models, and workload regimes.
>
> 2. Downstream risks:
> - We will add a concise discussion that improving long-horizon memory systems may increase privacy exposure and misuse risk if safeguards are weak.
> - We will clarify that privacy/deletion guarantees are not empirically validated in this paper.
> - We will add uncertainty reporting for key comparisons in the revision.

---

> > ### Author Rebuttal · Reviewer_pboU · 2026-04-02
> >
> > Thank you for your response, I will increase my score to 5.

---

### Official Review · Reviewer_826V · 2026-03-12

**Soundness:** 3
**Presentation:** 3
**Significance:** 3
**Originality:** 3
**Overall Recommendation:** 4
**Confidence:** 4

**Summary:**

This paper introduces Neuromem, a dedicated testbed for benchmarking External Memory Modules (EMMs) under a realistic, interleaved streaming protocol. Its core contribution is a five-dimensional lifecycle decomposition, enabling granular attribution of design choices to accuracy and latency performance.

**Compliance With Llm Reviewing Policy:**

Affirmed.

**Final Justification:**

As I have commented during Rebuttal Acknowledgement, I'd like to raise the rating to weak acceptance.

**Key Questions For Authors:**

1. The use of token-level F1 for evaluating generation in open-ended tasks may be brittle. Could performance drops be partly due to lexical, not semantic, mismatches? Did you consider more robust metrics like LLM-as-a-judge or semantic similarity?

2. The observed "latency tax" for generative steps is based on Pangu-1B and Llama-3-8B. Would this trade-off change if a highly optimized, sub-1B model were dedicated solely to memory operations?

3. Your interleaved protocol is described as blocking. Does the framework support asynchronous, non-blocking memory updates, or is the pipeline strictly synchronous during maintenance?

**Limitations:**

Some limitations are discussed in the paper. However, it would be highly beneficial to also see a deeper reflection on the potential constraints of relying on token-level F1 as the primary evaluation metric.

**Strengths And Weaknesses:**

Strengths:

1. The shift to a streaming, interleaved evaluation protocol is a practical contribution, as it mirrors real deployment conditions.

2. The five-dimensional lifecycle decomposition (D1-D5) effectively enables precise attribution of performance and latency to specific design choices.

Weaknesses:

1. The reliance on token-level F1 with Porter stemming is overly strict for evaluating LLM-generated text, as it penalizes valid paraphrases, potentially skewing accuracy comparisons.

2. The D1-D5 decomposition, while clear, abstracts complex systems into synchronous atomic steps. This risks overlooking the asynchronous, system-level optimizations that are key to the efficiency of agents like MemGPT.

3. The central finding of a generative "latency tax" may be influenced by the specific synchronous pipeline and hardware used. A discussion on how modern asynchronous serving might alter this trade-off would strengthen the claim. Furthermore, the exclusion of iterative graph architectures (e.g., HippoRAG) limits the scope of the design guidelines.

---

> ### Author Rebuttal · Authors · 2026-03-31
>
> Thank you for the detailed feedback. Your suggestions helped us improve both evaluation scope and claim precision.
>
> ## Response to Weaknesses
> > W1. Token-level F1 may be too strict
>
> We agree that token-level F1 alone can penalize valid paraphrases. To address this, we added a complementary LLM-as-a-Judge evaluation (DeepSeek V3.2) and analyzed mismatch patterns. Many low-F1 cases preserve semantic meaning.
>
> | Error Category | Proportion |
> |---|---|
> | Lexical Substitution | 72.5% |
> | Format Variation | 11.5% |
>
> We will revise the paper to present F1 and Judge as complementary metrics: F1 for exact fidelity, Judge for semantic equivalence, with uncertainty reporting on key gaps.
>
> > W2. D1-D5 decomposition may miss asynchronous optimizations
>
> We agree this concern is important. Our current decomposition is intentionally evaluated under a blocking online regime that prioritizes immediate consistency. We will clarify that this is a deliberate scope choice for controlled attribution, not a claim that asynchronous systems are invalid.
>
> > W3. Latency tax may depend on synchronous pipeline/hardware
>
> Agreed. We will narrow the wording to state that observed generative latency tax is measured under our current serving stack and blocking protocol. We will add discussion that asynchronous serving can hide wall-clock delay at the cost of consistency/visibility trade-offs.
>
> >  W4. Excluding iterative graph architectures (for example HippoRAG)
>
> We understand this concern and will clarify the fairness rationale. As detailed in our appendix discussion regarding HippoRAG, iterative graph methods exceeded practical latency budgets in our online blocking setup. After roughly 100 turns, a single insertion could take over 10 minutes. We therefore excluded them from the main online benchmark for scope consistency, and will explicitly frame this as a setting mismatch rather than a negative judgment of those methods. We will also disclose the exclusion threshold/protocol in the appendix.
>
> ## Response to Questions
>
> >  Q1. Could drops be lexical mismatch rather than semantic failure?
>
> Yes, and we appreciate this suggestion. Our new Judge analysis confirms substantial lexical mismatch effects. We also observed task dependence: divergence between Judge and F1 is larger in generative operations, while factual operations are more aligned. Importantly, performance ranking across strategies remains broadly consistent across both metrics.
>
> | Round | F1 Drop | Judge Drop | Ratio |
> |---|---|---|---|
> | R2 | 12.14% | 7.47% | 1.63x |
> | R3 | 16.21% | 11.72% | 1.38x |
> | R4 | 17.77% | 12.07% | 1.47x |
> | R5 | 23.81% | 16.88% | 1.41x |
>
> > Q2. Would a dedicated sub-1B model change the trade-off?
>
> Potentially yes. Our current evidence suggests latency tax is not strictly monotonic with parameter count; architecture and serving efficiency matter substantially. A specialized compact model could move the quality-latency frontier, but training and evaluating dedicated memory operators is beyond current scope. We will state this as a promising future direction and avoid absolute conclusions.
>
> > Q3. Does the framework support asynchronous non-blocking maintenance?
>
> The current benchmark protocol is synchronous/blocking by design to enforce immediate consistency in online evaluation. Neuromem as a framework is modular and can be extended to asynchronous operators, but our submitted results are intentionally reported under synchronous constraints for controlled, comparable attribution. We will add this as an explicit scope statement in the main text.
>
> ## Response to Limitations Comment
> We agree and will expand limitations on metric choice. Specifically, we will explicitly state that token-level F1 can underestimate semantically correct paraphrases, and we will include the complementary Judge results in the revision to provide a more balanced quality assessment.

---

> > ### Author Rebuttal · Reviewer_826V · 2026-04-03
> >
> > Thank you for the response. The supplementary LLM-as-a-judge evaluation provides helpful context for the performance rankings. Additionally, your clarifications on the synchronous blocking regime and the rationale for excluding specific graph architectures establish clearer boundaries for the evaluation. Therefore, I will raise my score.

---

### Official Review · Reviewer_7WJi · 2026-03-12

**Soundness:** 3
**Presentation:** 3
**Significance:** 3
**Originality:** 3
**Overall Recommendation:** 4
**Confidence:** 4

**Summary:**

This work appears to explore a central concept in the evaluation of external memory modules for LLM agents, namely the need to study memory performance under a realistic streaming regime where insertion and retrieval are interleaved.

**Compliance With Llm Reviewing Policy:**

Affirmed.

**Key Questions For Authors:**

I don't have any questions.

**Limitations:**

No they don't have a limitation section.

**Strengths And Weaknesses:**

## Strengths

1. The paper addresses an important and timely systems question regarding realistic evaluation of long-term memory in LLM agents.
2. The lifecycle decomposition provides a useful conceptual framework for analyzing design trade-offs.
3. Experiments span several representative benchmarks, improving empirical coverage compared to many prior works.

## Weaknesses

1. **Novelty**: The primary contribution is largely empirical benchmarking and analytical decomposition, with limited methodological novelty in memory modeling or learning algorithms.
2. **Evaluation metric scope is somewhat narrow.**: Token-level F1 provides a convenient proxy for reasoning accuracy, but may not fully capture higher-level agent performance aspects such as task completion success, personalization quality, or long-term planning consistency.

---

> ### Author Rebuttal · Authors · 2026-03-31
>
> Thank you for the positive and constructive assessment.
>
> ## Response to Weaknesses
>
> > W1. Limited novelty in memory modeling/learning algorithms
>
> We agree with this characterization and will sharpen our positioning accordingly. Neuromem is not presented as a new memory-learning algorithm. Its core contribution is methodological: a lifecycle-factorized, reproducible evaluation framework that enables causal attribution of quality-latency behavior in realistic streaming settings.
>
> > W2. Metric scope is narrow
>
> We agree that higher level agent metrics are important. In this submission, we intentionally focused on infrastructure level diagnostics such as token level fidelity and latency to isolate memory pipeline effects from broader agent orchestration confounders.
>
> In our revision, we will add complementary semantic evaluations using an LLM Judge and temporal consistency reporting. Preliminary evaluations on LoCoMo time related queries demonstrate this approach:
>
> | Strategy | Token F1 | LLM Judge |
> |---|---|---|
> | Inverted Vector | 0.1260 | 0.0824 |
> | Queue Segment | 0.1126 | 0.0785 |
>
> These results reveal that the LLM Judge applies stricter semantic penalties to temporal queries than standard token matching. We also cross validated this temporal reasoning stability on a subset of LongMemEval. Finally, we will clarify that end task success, personalization, and planning remain prioritized future extensions.
>
> ## Response to Questions
> No explicit questions were raised. We appreciate the clear feedback and have incorporated the requested clarifications into our revised framing.
>
> ## Response to Limitations Comment
> Thank you for pointing this out. A limitations section is present in the submission, but we agree visibility can be improved. We will make the section heading more prominent, move scope language earlier in the conclusion, and expand limitations with clearer claim boundaries and downstream risk discussion.

---

> > ### Author Rebuttal · Reviewer_7WJi · 2026-04-04
> >
> > Thank the authors for the response. I will maintain my positive score.

---

### Official Review · Reviewer_jCQY · 2026-03-16

**Soundness:** 3
**Presentation:** 3
**Significance:** 3
**Originality:** 3
**Overall Recommendation:** 4
**Confidence:** 4

**Summary:**

This work appears to explore a central concept in LLM systems evaluation: how to benchmark external memory modules under realistic streaming conditions, where insertions and retrievals interleave over time rather than occurring in a static build-then-query regime. Overall, this paper presents a central concept that many existing memory benchmarks miss the true deployment setting because they do not decompose the full memory lifecycle or attribute accuracy/latency tradeoffs to specific design choices. The paper introduces Neuromem, a unified testbed that factorizes external memory systems into five dimensions—data structure, normalization, consolidation, query formulation, and context integration—and evaluates interchangeable variants under a chronological streaming protocol on LoCoMo, LongMemEval, and MemoryAgentBench.

The main empirical claim is that memory performance generally degrades as history grows, the memory data structure determines the achievable quality frontier, and many generative interventions mainly add latency while offering little accuracy improvement. The paper argues for lifecycle-aware, streaming evaluation as a more informative basis for system design.

**Compliance With Llm Reviewing Policy:**

Affirmed.

**Final Justification:**

My concerns regarding the manuscript are fully resolved, according to authors’ responses and explanations.

**Key Questions For Authors:**

1. How robust are the conclusions about “generative latency tax” across stronger or more recent models? The paper includes some robustness checks, but the current claims still feel broader than the evidence.
2. Can the authors provide more evidence that the re-instantiated representative systems preserve the essential behavior of the original methods, rather than introducing artifacts from the shared stack abstraction?
3. Why is token-level F1 the right primary metric here? Could the paper include retrieval quality, citation accuracy, temporal consistency, or end-task success metrics?
4. The paper argues that data structure sets the accuracy ceiling. How much of that conclusion depends on the specific hyperparameter choices for D4/D5, and how sensitive is it to better tuning of the weaker structures?
5. Could the authors extend D2–D5 ablations beyond LoCoMo, even on smaller subsets, to show that the same trends hold across workloads?

**Limitations:**

The paper would be stronger with a more careful claim scope. I would tone down universal statements like “semantic compression is lossy” or “generative optimization is inefficient” and instead frame them as empirical findings under the tested stacks and workloads.
I would also add one section explicitly discussing fidelity vs abstraction in the decomposition: what is preserved from the original systems, what is lost, and how this affects interpretation.
A richer metric suite would improve the paper. Since the submission is about evaluating memory systems under realistic streaming use, evaluation should go beyond token-level F1 and latency alone.
Finally, a small-scale multi-dataset validation for D2–D5 would help a lot. Even partial confirmation outside LoCoMo would strengthen confidence that these findings are not benchmark-specific.

**Strengths And Weaknesses:**

## Strengths

The paper is well motivated. The critique of static evaluation is convincing: many deployed memory systems do operate in an interleaved insert/retrieve setting, and measuring only final-state performance obscures important system dynamics. The lifecycle decomposition into five dimensions is intuitive and useful as an organizing framework.

The paper’s experimental framing is one of its biggest strengths. Instead of treating each memory method as a monolithic black box, it maps representative systems into a shared D1–D5 taxonomy and evaluates components under a unified serving stack. That makes the empirical conclusions more actionable than a standard leaderboard paper.

I also appreciate the practical emphasis on latency attribution, not just answer quality. The distinction between insertion latency and retrieval latency is important, especially for real-time agent deployments. The observation that some methods merely shift computational debt between insertion and retrieval rather than reducing total cost is a useful systems insight.

Several of the findings are interesting and plausible. The paper shows that hybrid or multi-layer storage structures tend to dominate the reasoning frontier, while aggressive semantic compression or generative query/context operations often degrade cost-effectiveness. These are valuable design lessons for practitioners building long-horizon memory systems.

The paper is also commendably explicit about limitations, especially the computational intractability of some graph-refinement paradigms and the scarcity of datasets that support granular state verification over time.

## Weaknesses / concerns

My main concern is novelty depth. The paper’s strongest contribution is an evaluation framework and empirical decomposition, not a new memory algorithm. That is still worthwhile, but for ICML the contribution may feel more benchmarking/systems-oriented than methodologically novel. The paper should sharpen this positioning and be explicit that its value lies in evaluation methodology and experimental attribution.

A second concern is the strength of some of the paper’s conclusions. Several findings are phrased quite categorically—for example that semantic compression is destructive or that generative optimization is a latency trap—but the evidence may not fully support such broad generalization beyond the chosen implementations, workloads, and backbone models. In multiple places the paper risks overclaiming from a relatively bounded experimental scope.

Third, the evaluation, while thoughtful, is still somewhat limited in coverage and fairness. The full D1–D5 ablation is concentrated on LoCoMo, with the other datasets mainly used for cross-validating D1 trends. That is understandable given combinatorial cost, but it weakens the universality of conclusions about normalization, consolidation, and integration.

I am also somewhat concerned about baseline completeness and implementation fidelity. The paper decomposes many prior systems into shared operators, which is useful, but this abstraction may under-represent synergies within the original full systems. The empirical results then partly evaluate the authors’ re-instantiations rather than the systems as originally designed. The paper acknowledges implementation details in the appendix, but this remains an important caveat.

Another concern is that the paper uses token-level F1 as the main reasoning metric. That is practical, but for conversational memory tasks it may miss whether retrieved evidence is semantically correct, temporally grounded, or useful for downstream generation. Since the paper argues for lifecycle-aware realism, I would have liked richer task-level quality measures alongside F1.

Finally, some of the most interesting claims—such as privacy compliance benefits of heuristic maintenance—are more suggestive than demonstrated. The impact statement raises important ideas, but the paper does not empirically evaluate privacy or deletion guarantees. Those parts should be framed more cautiously.

---

> ### Author Rebuttal · Authors · 2026-03-31
>
> Thank you for the detailed and actionable feedback. We appreciate your focus on claim scope, fidelity, and evaluation completeness.
>
> ## Response to Weaknesses
>
> > W1. Contribution should be positioned as evaluation methodology
>
> Agreed. We will explicitly position Neuromem as an evaluation and attribution framework rather than a new memory algorithm. We will emphasize methodological novelty in lifecycle decomposition (D1-D5), controlled benchmarking, and causal attribution of quality-latency trade-offs.
>
> > W2. Some conclusions are too categorical
>
> Agreed. We will replace broad wording with scoped empirical wording tied to tested stacks/workloads. Our revised language will avoid universal claims and state deployment assumptions explicitly.
>
> > W3. Coverage beyond LoCoMo is limited
>
> You are right. Full D1-D5 ablation is concentrated on LoCoMo due to combinatorial cost. To strengthen cross-workload confidence, we ran additional subset validation on LongMemEval and observed directionally consistent trends. We will report this as partial validation, not full universality evidence.
>
> > W4. Concern about baseline fidelity under shared abstraction
>
> We agree this needs stronger documentation. In revision, we will add explicit fidelity reporting for representative methods (original vs reinstantiated deltas under matched data splits, prompts, and latency budgets), plus a clear preservation/pruning map to show what is retained and what is simplified by the shared stack. We will also release the validation protocol and run logs.
>
> > W5. Token-level F1 alone is insufficient
>
> Agreed. We will keep F1 as an exact-fidelity signal and add complementary metrics for semantic and temporal quality. Specifically, we include LLM-as-a-Judge comparison and dataset-native temporal consistency analysis.
>
> > W6. Privacy/deletion claims in impact statement are suggestive
>
> Agreed. We will tone these statements down and explicitly label them as theoretical implications rather than empirically validated guarantees.
>
> ## Response to Questions
>
> > Q1. Robustness of latency-tax conclusions on stronger/newer models
>
> Our claim is scoped to current deployment constraints, not to all future models. In our measured online blocking regime, generative maintenance remains substantially slower than lightweight heuristics. We will explicitly note that stronger models and future systems may shift this frontier.
>
> > Q2. Evidence that reinstantiated systems preserve essential behavior
>
> We will provide stronger evidence in the revision: method-level fidelity tables with original-vs-reinstantiated performance deltas, matched protocol details, and explicit notes for components that cannot be faithfully reproduced under online blocking constraints.
>
> > Q3. Why use token-level F1 as primary metric?
>
> We use F1 as the primary metric for exact factual fidelity under memory perturbation. We agree this is not sufficient alone, so we now add complementary semantic and temporal views (LLM Judge and temporal consistency metrics). For end-task success in full agent loops, we will clearly mark this as future work beyond the current infrastructure-focused scope.
>
> > Q4. Sensitivity of structural-ceiling claims to D4/D5 tuning
>
> We will include additional sensitivity analysis under fixed latency budgets with confidence intervals. Our current observation is that tuning can improve methods locally, but does not reverse structural ranking under the tested online regime. We will present this as empirical evidence, not a universal theorem.
>
> > Q5. Extend D2-D5 ablations beyond LoCoMo
>
> We performed subset-scale extensions on LongMemEval and observed consistent directional behavior.
>
> | Ingestion Strategy | F1 Score | Execution Latency | Relative Delay |
> |---|---|---|---|
> | Queue Segment Base | 12.5% | 114ms | Baseline |
> | Plus Enrich | 11.9% | 709ms | 522% penalty |
> | Plus Rewrite | 8.3% | 506ms | 344% penalty |
>
> We will include these as partial cross-workload evidence and explicitly state remaining uncertainty for full generalization.
>
> ## Response to Limitations Suggestions
> We agree with all three suggestions and will revise accordingly.
>
> 1. Claim scope:
> - Replace universal statements with condition-bounded empirical claims.
>
> 2. Fidelity vs abstraction:
> - Add a dedicated subsection on what is preserved, what is abstracted, and interpretation implications.
>
> 3. Richer metric suite and broader validation:
> - Add complementary semantic/temporal metrics and include subset multi-dataset D2-D5 validation.
> - Add uncertainty reporting (for example confidence intervals) for key comparisons.

---

> > ### Author Rebuttal · Reviewer_jCQY · 2026-04-03
> >
> > My concerns regarding the manuscript are fully resolved, according to authors’ responses and explanations.

---

### Decision · Program_Chairs · 2026-04-30

**Decision:**

Accept (regular)

**Comment:**

The paper proposes Neuromem, a lifecycle-factorized evaluation framework for external memory modules under realistic streaming conditions. Reviewers agreed the streaming evaluation paradigm is well-motivated and the D1-D5 decomposition provides actionable design attribution that goes beyond standard leaderboard comparisons. The main concerns were overclaiming beyond experimental scope, reliance on token-level F1 as the sole quality metric, and limited cross-dataset coverage for D2-D5 ablations. The rebuttal addressed these with new LLM-as-Judge evaluations, partial cross-dataset validation on LongMemEval, and revised claim scoping. Cross-dataset generalization evidence remains subset-scale, and the authors should ensure the promised fidelity tables and expanded metrics are included in the camera-ready. The paper meets the acceptance bar as a practical evaluation and design-study contribution to the LLM memory systems area.